# AN1-type zinc finger protein 3 (ZFAND3) is a transcriptional regulator that drives Glioblastoma invasion

Anne Schuster[1], Eliane Klein[1], Virginie Neirinckx [1], Arnon Møldrup Knudsen [2,3], Carina Fabian[1,4], Ann-Christin Hau [1], Monika Dieterle [1], Anais Oudin[1], Petr V. Nazarov [5], Anna Golebiewska [1], Arnaud Muller[5], Daniel Perez-Hernandez[5], Sophie Rodius[5], Gunnar Dittmar [5], Rolf Bjerkvig [1,4], Christel Herold-Mende [6], Barbara Klink [7,8], Bjarne Winther Kristensen[2,3] & Simone P. Niclou [1,4✉]

The infiltrative nature of Glioblastoma (GBM), the most aggressive primary brain tumor, critically prevents complete surgical resection and masks tumor cells behind the blood brain barrier reducing the efficacy of systemic treatment. Here, we use a genome-wide interference screen to determine invasion-essential genes and identify the AN1/A20 zinc finger domain containing protein 3 (ZFAND3) as a crucial driver of GBM invasion. Using patient-derived cellular models, we show that loss of ZFAND3 hampers the invasive capacity of GBM, whereas ZFAND3 overexpression increases motility in cells that were initially not invasive. At the mechanistic level, we find that ZFAND3 activity requires nuclear localization and integral zinc-finger domains. Our findings indicate that ZFAND3 acts within a nuclear protein complex to activate gene transcription and regulates the promoter of invasion-related genes such as COL6A2, FN1, and NRCAM. Further investigation in ZFAND3 function in GBM and other invasive cancers is warranted.

[1] NORLUX Neuro-Oncology Laboratory, Department of Oncology, Luxembourg Institute of Health, Luxembourg, Luxembourg. [2] Department of Pathology, Odense University Hospital, Odense, Denmark. [3] Department of Clinical Research, University of Southern Denmark, Odense, Denmark. [4] Department of Biomedicine, University of Bergen, Bergen, Norway. [5] Quantitative Biology Unit, Luxembourg Institute of Health, Luxembourg, Luxembourg. [6] Division of Neurosurgical Research, Department of Neurosurgery, University of Heidelberg, Heidelberg, Germany. [7] National Center of Genetics, Laboratoire National de Santé, Dudelange, Luxembourg. [8] Functional Tumor Genetics, Department of Oncology, Luxembourg Institute of Health, Luxembourg, Luxembourg. ✉email: simone.niclou@lih.lu

Cancer cell invasion and ensuing metastasis are a leading cause of death. Malignant tumors of the brain, including Glioblastoma (GBM), are characterized by a high invasive capacity leading to a spread throughout the brain parenchyma[1], a growth pattern which is largely accountable for the current therapeutic failure and poor patient outcome. Invasive cells that migrate away from the tumor core escape surgical resection, are partially sheltered from radio- and chemotherapy and are not detected by standard imaging techniques. Furthermore it was recently shown that glial tumors form multicellular networks through ultra-long membrane protrusions, so-called tumor microtubes, that facilitate brain invasion and contribute to treatment resistance[2,3].

Due to the specific structure of the adult brain, GBM invasion differs from vascular or lymphathic pathways classically associated with peripheral metastatic cancer. GBM cells insinuate themselves in the interstitial space of the neural tissue or migrate along blood vessels and white matter tracts, relying on basal membranes and extracellular matrix (ECM) components[4]. Although significant efforts were carried out to elucidate the mechanisms underlying GBM cell invasion (e.g., cytoskeleton remodeling, secretion of proteases, intracellular signaling)[5], therapeutic approaches targeting GBM invasion have not heralded any benefit so far and novel targets regulating the invasive process are actively being pursued[6–8].

RNA interference screens are powerful tools to uncover gene function and their contribution to specific cellular phenotypes[9]. Such approaches allowed e.g., the identification of genes involved in cell migration and invasion in various cancer models[10,11] including GBM[12]. Here, we applied genome-wide RNA interference in GBM and identified AN1-Type Zinc Finger protein 3 (ZFAND3) as a key regulator of GBM cell invasion. Zinc finger (ZF) proteins are involved in nucleic acid recognition, transcriptional activation, protein folding and assembly, however the function of ZFAND3 remains unknown. *ZFAND3* (also known as testis expressed sequence 27, *Tex27*) was initially characterized during mouse sperm maturation[13,14] and was associated with susceptibility for development of type 2 diabetes in humans[15,16]. We find that ZFAND3 strongly potentiates invasiveness of GBM patient-derived cells in vitro, ex vivo and in vivo. We show that nuclear *ZFAND3* expression is increased in the infiltrative compartment in GBM patient biopsies and that nuclear localization is essential for ZFAND3 activity. Finally we identify ZFAND3 as a transcription factor that regulates expression of adhesion and invasion-related genes.

## Results

### RNA interference screen identifies *ZFAND3* as a candidate gene involved in GBM invasion.

Although GBMs in patients are invariably invasive, not all patient-derived GBM cells display the same invasion capacity. Inter-patient differences can be observed when patient-derived GBM stem-like cells (GSCs) are implanted into the mouse brain: Non-invasive (NI) cells grow as circumscribed tumors displaying aberrant blood vessels and necrosis; low invasive (LI) cells partially invade into the cortex and traverse the corpus callosum to the contralateral hemisphere and highly invasive cells (HI) completely colonize the brain parenchyma of both hemispheres (Fig. 1a, additional examples on Supplementary Fig. 1a, b). We have previously described similar histological phenotypes with differential invasive potential in GBM patient-derived orthotopic xenografts based on organotypic tumor spheroids[17]. Interestingly the distinct invasive behavior of GSCs could be recapitulated in vitro, e.g., using 3D-Boyden chamber assays (Fig. 1b, c, Supplementary Fig. 1a, b) or sphere sprouting assays (Fig. 1d, e). We further confirmed the differential invasive

phenotypes in ex vivo invasion assays in organotypic brain slice cultures, which allowed to determine differences in single cell velocity (Fig. 1f–i). The invasion capacity was correlated to the expression of some (*CDH2, MMP2, SNAI1, ZEB1*) but not all known invasion markers in vitro and in vivo (Supplementary Fig. 1c–h), but did not correlate to transcriptional GBM subtypes (proneural, mesenchymal, classical) as defined by Wang et al.[18] (Supplementary Fig. 1i). Taken together these data indicate that patient-derived GSCs faithfully reflect the heterogeneity and invasion capacity of GBM in vivo, ex vivo and in vitro.

Using highly invasive (HI) GBM GSCs, we performed a genome-wide loss-of-function shRNA screen to uncover novel key drivers of GBM invasion. Invasion-defective and invasion-competent cells were specifically isolated using the Boyden chamber assay (Fig. 2a). In highly aneuploid cancer cells, such as GBM GSCs, RNA interference may be more reliable then CRISPR based knockout screens, because the transcripts rather than the DNA are targeted[9]. Focusing on the invasion-defective cells, barcoded shRNAs were sequenced and a stringent bioinformatic analysis pipeline was applied by combining four of the most common analysis methods: RSA[19], RIGER[20], MAGeCK[21,22], and HiTSelect[23] (Supplementary Fig. 2a, b). Only the gene candidates in common between all four methods were selected, resulting in a set of 17 invasion-essential candidate genes within the 2% top hits (Fig. 2b). This included Colony stimulating factor 1 (*CSF1*), a known cytokine involved in invasion and metastasis. To further reduce the number of gene candidates of interest, we analysed the expression of the 17 genes in NI, LI, and HI cells in vitro and when grown as xenografts in vivo. Compared to other candidate genes, the AN1-Type Zinc Finger protein 3 (*ZFAND3*) gene showed higher expression in HI cells compared to NI and LI cells, in vitro as well as in vivo, and corresponding knockdown clones were enriched in the analysis (Supplementary Fig. 2b–d), we therefore focused on *ZFAND3* for further analysis. Quantitative real time PCR (qPCR) confirmed a higher expression of *ZFAND3* in HI cells, compared to LI and NI (Fig. 2c). Immunohistochemistry of corresponding GBM xenografts in the mouse brain showed more ZFAND3 positive cells in HI tumors in comparison to LI tumors (Fig. 2d). Analysis of TCGA data via the GlioVis platform[24] revealed strong *ZFAND3* expression in all classes of diffuse glioma compared to nontumor controls (Supplementary Fig. 2e), in line with the high invasive potential of these tumors. There was no correlation with transcriptional GBM subtypes (Supplementary Fig. 2f). Pan-tumor studies based on TCGA and GTEx databases (via GEPIA platform)[25,26] also highlighted increased *ZFAND3* expression in diverse cancers compared to control tissue, in particular in pancreatic adenocarcinoma and melanoma (Supplementary Fig. 2g). Next we analysed ZFAND3 protein expression in 21 fresh GBM patient biopsies. ZFAND3 protein was detected in the majority of GBM (17/21) and throughout different tumor compartments, including central, intermediate and peripheral areas (Fig. 2e). By immunofluorescence, we found that ZFAND3 was present in the cytoplasm and the nucleus of GBM cells (identified by P53 staining) and we analysed the fraction of ZFAND3 positive tumor cells based on subcellular localization (Fig. 2f and Supplementary Fig. 3a, b). The fraction of positive cells was similar throughout different tumor compartments, both for cytoplasmic and nuclear staining (Supplementary Fig. 3c, d). However, in contrast to cytoplasmic ZFAND3 staining (Fig. 2g), we found that both the staining intensity and the ratio of nuclear/cytoplasmic ZFAND3 staining were increased in peripheral tumor cells compared to central cells, indicating that the relative fraction of tumor cells with nuclear ZFAND3 is higher in the tumor periphery, and that nuclear ZFAND3 is expressed to a higher extent in these cells (Fig. 2h, i). Since ZFAND3 appeared also in nontumor cells, we

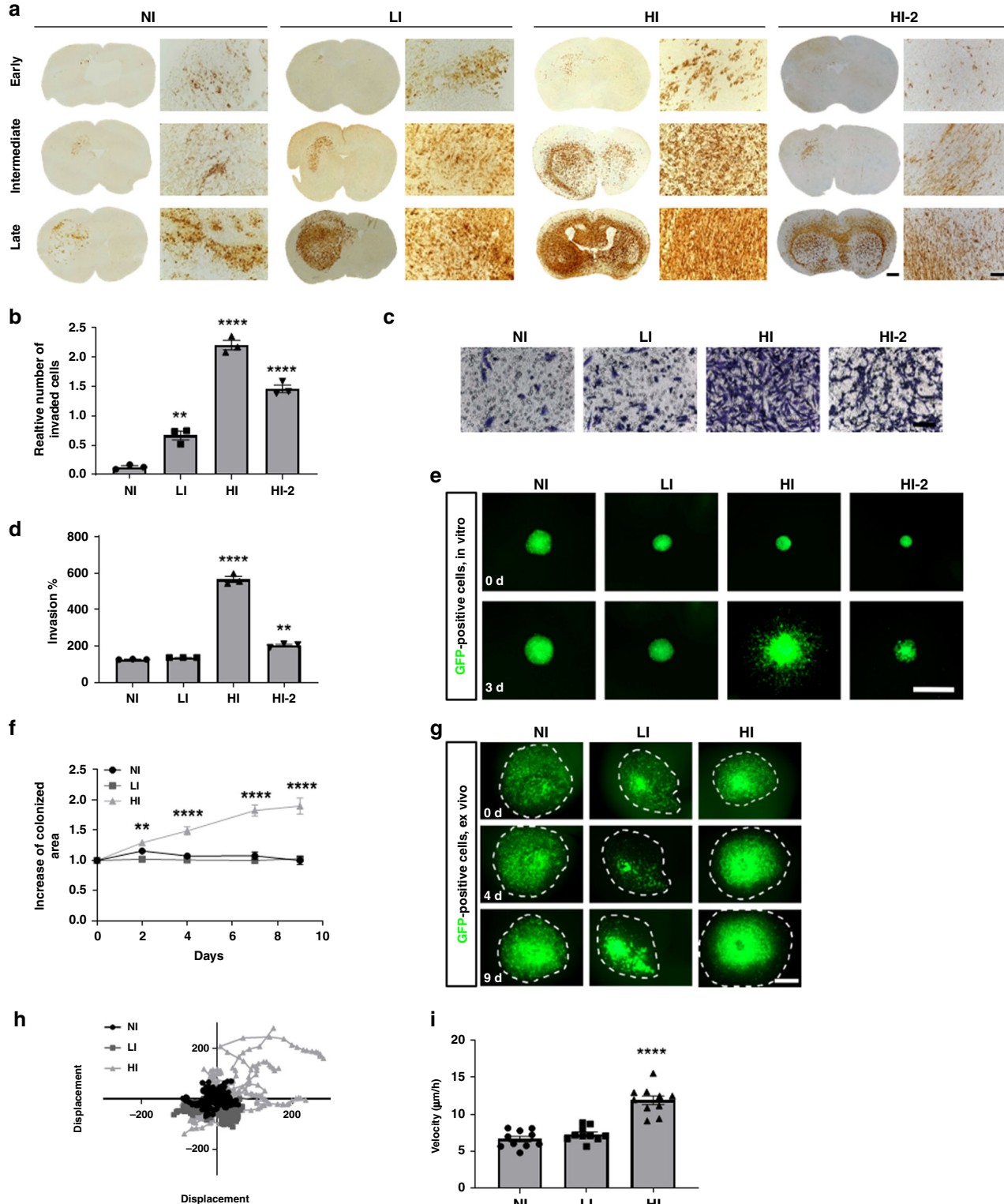

**Fig. 1 Patient-derived GBM stem-like cells (GSCs) display different invasion phenotypes in vivo, which are recapitulated in in vitro and ex vivo assays.**
**a** Early, intermediate and late time point of different GSC orthotopic xenografts in mice displaying non invasive (NI), low invasive (LI) and highly invasive (HI, HI-2) phenotypes. Respective tumor development times were 5 weeks (NI), 8 weeks (LI) and 25 weeks (HI, HI-2). Anti-human vimentin staining was used to visualize tumor cells (Scale bars = 100 μm and 1000 μm for overview). ($n = 2$ mice for each tumor and time point with $n = 3$-4 sections per mouse). **b** In vitro Boyden Chamber invasion assay, displaying the relative number of invaded cells ($n = 3$). **c** Representative pictures of in vitro Boyden chamber invasion assay reflecting the different invasion phenotypes. Scale bar 100 μm. **d** Quantification of invasion of GSCs in 3D sprouting assay ($n = 3$). **e** Representative pictures of sprouting assay of different GFP-positive GSCs ($n = 3$) (Scale bar = 1000 μm). GFP: Green fluorescent protein. **f** Quantification of increase of colonized area of GSCs in ex vivo brain slice cultures, 9 days after tumor implantation ($n = 10$). **g** Representative pictures of GSCs in ex vivo brain slice cultures at day 0, 4, and 9 (Scale bar = 1000 μm). **h** Displacement of GSCs injected into ex vivo brain slice cultures ($n = 10$). **i** Velocity of GSCs in ex vivo brain slice cultures. Results of (**b**, **d**, **f** are **i**) displayed as average ± SEM and were analyzed with one-way ANOVA. **$p_{value} < 0.01$, ****$p_{value} < 0.0001$.

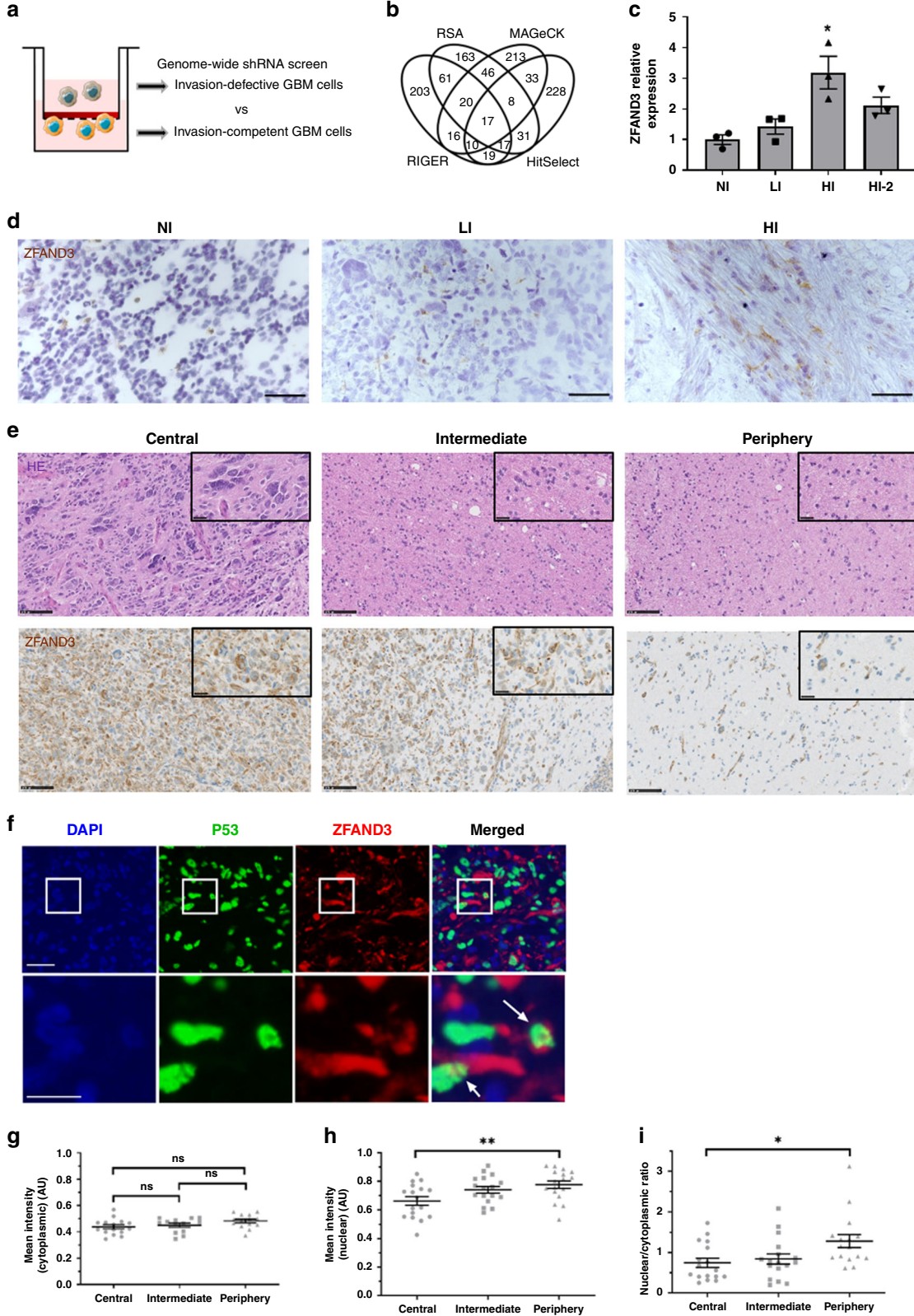

performed colabeling for Iba1 (microglia marker) and NeuN (neuronal marker). We observed some colocalization with Iba1, but not with NeuN, both in clinical samples (Supplementary Fig. 3e) and in xenografts (Supplementary Fig. 3f), indicating that ZFAND3 is expressed by a subpopulation of microglial cells. The majority of staining was found in the tumor area rather than in the neuropil (Supplementary Fig. 3f lower panel). In conclusion,

we identified *ZFAND3* as an invasion-related gene that displays increased nuclear expression in the infiltrative tumor compartment of clinical GBM specimen.

**ZFAND3 knockdown impairs GBM cell invasion in vitro, ex vivo and in vivo.** To confirm the functional screen data,

**Fig. 2 Expression of ZFAND3 is associated with GBM cell invasion. a** Set up of genome-wide shRNA pooled screen selecting for invasion-defective and invasion-competent cells in ECM-collagen coated transwell chambers. **b** Loss-of-function screen focused on shRNAs significantly enriched in invasion-defective cells. Results were analysed with RSA, RIGER, MAGeCK and HitSelect methods, identifying 17 genes as the top 2% common hits. **c** qPCR showed higher *ZFAND3* expression in highly invasive GSCs (HI, HI-2) compared to non invasive (NI) ($n = 3$; $p = 0.007$) and low invasive (LI) cells ($n = 3$ biologically independent samples). Results are displayed as average ± SEM and were anlysed with one-way ANOVA. *$p_{value}$ < 0.1. **d** IHC for ZFAND3 in intracranial GBM xenografts in mice generated from respected patient-derived GSCs (NI, LI HI) (Scale bars = 50 μm, hematoxylin counterstaining). **e** IHC revealed ZFAND3 protein in GBM patient biopsies in the tumor core (central), intermediate area (intermediate) and tumor margin with diffuse infiltration (periphery) (Scale bars = 100 μm). Representative images are shown ($n = 21$). **f** Triple immunofluorescence identified ZFAND3 staining in cytoplasm and nucleus of positive tumor cells: ZFAND3 (red), P53 (green) and DAPI (blue) (Scale bar upper row = 50 μm, lower row = 15 μm). ($n = 17$ different partient samples). Mean intensity of cytoplasmic (**g**) and nuclear (**h**) ZFAND3+ tumor cells in central, intermediate, and peripheral tumor areas, the latter being increased in the periphery ($n = 17$). Data were analyzed as matched data with one-way ANOVA and Tukey´s multiple comparison test. ($p = 0.009$) **i** Ratio of nuclear/cytoplasmic ZFAND3 staining is increased in the periphery compared to the central tumor area ($n = 17$ patient samples). Data were analysed as matched data with Kruskal–Wallis test and Dunn's multiple comparison test ($p = 0.012$). *$p_{value}$ < 0.05, **$p_{value}$ < 0.01.

shRNA-mediated knockdown (KD) of *ZFAND3* was performed in two highly invasive GSCs (HI and HI-2). Efficient KD with two different shRNAs was achieved at RNA and protein level (Fig. 3a, b and Supplementary Fig. 4a–e). Similar to patient samples, endogenous ZFAND3 protein displayed cytoplasmic and nuclear localization, while KD cells only retained minor cytoplasmic staining (Fig. 3c). *ZFAND3* KD had no significant impact on cell proliferation (Fig. 3d). Using Boyden chamber assays on the same cells, *ZFAND3* KD significantly reduced invasion compared to shCtrl in HI cells (Fig. 3e, f). This was confirmed in the HI-2 cell line (Supplementary Fig. 4a–e).

To better imitate invasion in a brain microenvironment, we implanted GBM HI cells into ex vivo brain slices. In *ZFAND3* KD cells the area of colonization of the brain slice was decreased in contrast to shCtrl (Fig. 3g, h) and cellular velocity, as determined by single cell tracking, was reduced accordingly (Fig. 3i, j). Finally, we evaluated the invasion potential of *ZFAND3* KD cell in vivo: eight weeks after intracranial tumor implantation, the mice were sacrified and cell invasion to the contralateral hemisphere was quantified. We found significantly less cells in *ZFAND3* KD tumors compared to control (Fig. 3k, l and Supplementary Fig. 4j). Taken together these data validate the result of the screen and indicate that loss of ZFAND3 strongly impairs GBM cell invasion in vitro, ex vivo and in vivo.

**Expression of *ZFAND3* confers invasion potential to non-invasive patient-derived GBM cells.** Given that ZFAND3 downregulation considerably decreased the invasion potential of invasive GSCs, we asked if *ZFAND3* overexpression was able to bestow invasion capacity to non-invasive (NI) GSCs. We therefore expressed *ZFAND3* in NI GSCs, as shown by qPCR (Fig. 4a) and western blot analysis (Fig. 4b). Upon overexpression, ZFAND3 protein mostly accumulated in the nucleus (Fig. 4c) and did not affect proliferation of the cells (Fig. 4d). Instead we found that ZFAND3 increased invasion in vitro (Fig. 4e, f) and ex vivo in brain slice cultures. ZFAND3 expressing cells colonized a larger area (Fig. 4g, h) and displayed higher velocity (Fig. 4i, j). Upon transplantation in the mouse brain, ZFAND3 expressing tumors lost the circumscribed growth pattern of control NI cells (Fig. 4k and Supplementary Fig. 4k). The number of cells escaping the tumor mass was significantly increased for ZFAND3 expressing cells, compared to controls (Fig. 4l). These data indicate that ZFAND3 expression confers invasion potential to GBM cells that were initially not invasive.

**Nuclear localization of ZFAND3 is required for GBM cell invasion.** ZFAND3 contains two ZF domains, a N-terminal A20 domain and a C-terminal AN1 domain separated by a linker region (Fig. 5a). As we found ZFAND3 immunostaining in the nucleus and nuclear localization was increased in the infiltrative tumor compartment, we asked whether nuclear localization was needed for ZFAND3 activity. A ZFAND3 construct with a mutated nuclear localization signal (NLS) (ZFAND3-mutNLS) was expressed in NI cells (Fig. 5a–c and Supplementary Fig. 5a–c). This resulted in ZFAND3 accumulation in the cytoplasm (Fig. 5d) and prevented ZFAND3-induced invasion (Fig. 5f, g). The addition of a second NLS sequence to the mutant construct rescued both nuclear localisation and the invasion phenotype (Fig. 5a–g), indicating that ZFAND3 was active in the nucleus. No impact on proliferation was observed (Fig. 5e). In summary, these data demonstrate that nuclear localization is required for ZFAND3-induced invasion.

**Deletion of zinc-finger domains leads to loss of invasion phenotype.** In an attempt to further uncover the molecular basis of its activity, we generated ZFAND3 mutants with deletion of individual or both ZF domains (ZFAND3-Δ1, Δ2, and Δ1Δ2) (Fig. 5a and Supplementary Fig. 5a), and expressed them in NI GSCs (Fig. 5h, i). None of the variants exhibited proliferation defects (Fig. 5j). In contrast to full length ZFAND3, no increase in invasion was observed in cells expressing ZFAND3-Δ1, ZFAND3-Δ2 or ZFAND3-Δ1Δ2 (Fig. 5k, l), suggesting that both ZF domains are required for induction of invasion. Of note, while the double deletion construct accumulated in the cytoplasm, ZFAND3-Δ1 and ZFAND3-Δ2 correctly translocated to the nucleus (Fig. 5m). To further pinpoint the active residues, we generated point mutations in putative zinc-complexing amino acids (M1, M2, and M1-M2) (Supplementary Fig. 5a). These constructs retained nuclear localization, but also full activity with regard to the invasive phenotype (Supplementary Fig. 5b–g). In conclusion, although the exact residues conferring nuclear ZFAND3 activity remain elusive, our data hint to the need of both ZF domains to trigger invasion in GBM cells, suggesting that ZFAND3 may act as a transcriptional regulator.

**ZFAND3 is involved in transcriptional regulation of invasion-related genes.** To address whether ZFAND3 correlates with the expression of genes involved in migration and epithelial-mesenchymal transition, we analysed expression of *CDH2* (coding for N-cadherin), *MMP2*, *SNAI2* and *ZEB1* in GSCs with various ZFAND3 expression levels. Surprisingly, the modulation of *ZFAND3* expression (KD or overexpression) or function (mutation constructs) did not affect expression of these genes (Supplementary Fig. 6). We therefore performed RNA sequencing on shCtrl, shZFAND3-1 and shZFAND3-2 HI cells to gain broader insight into the transcriptional landscape upon *ZFAND3* knockdown. Among the differentially expressed genes (DEGs), 58 genes were significantly downregulated in both shZFAND3-1 and shZFAND3-2 versus

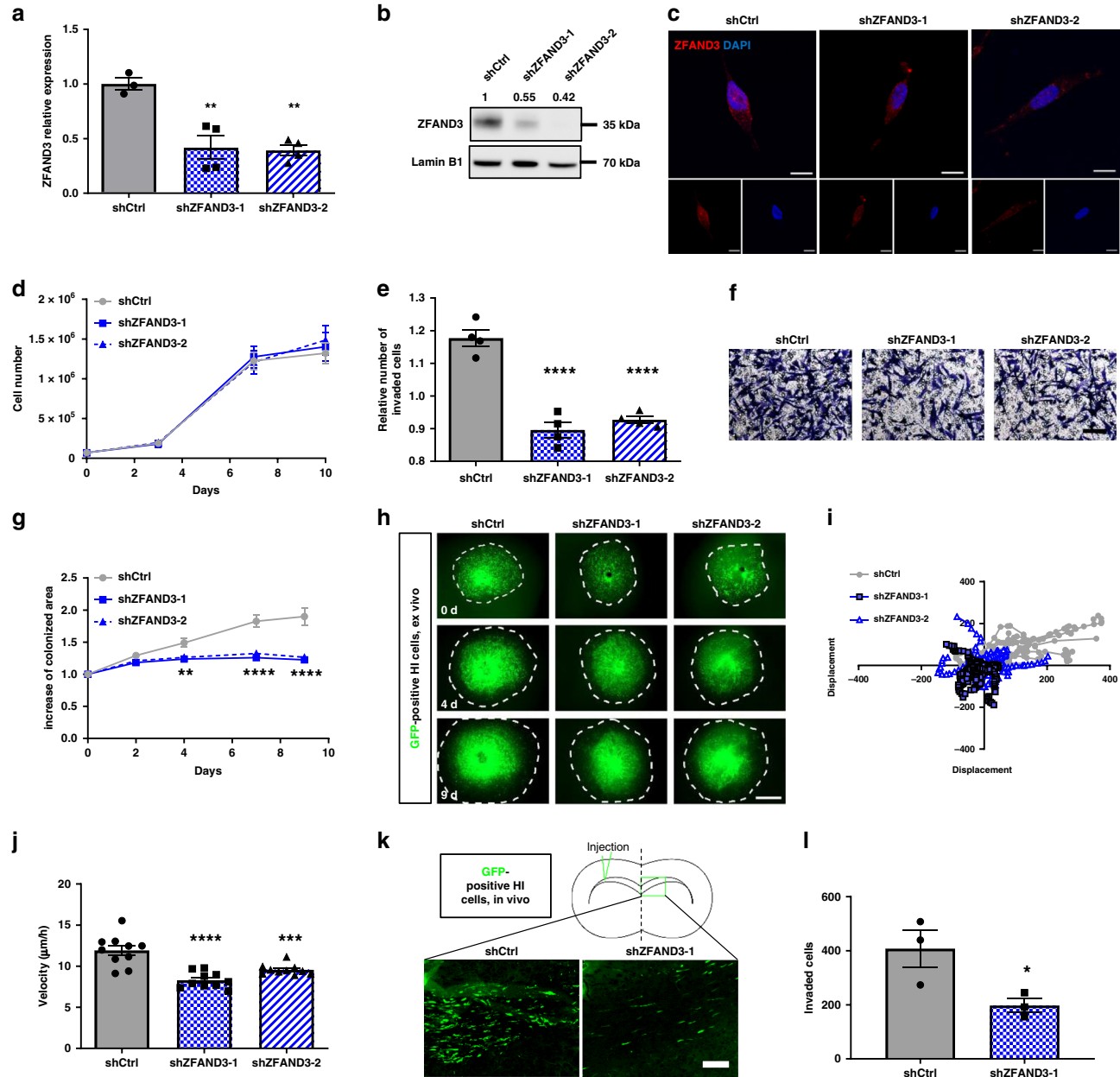

**Fig. 3 Knockdown of ZFAND3 decreases invasion capacity of highly-invasive GBM cells. a** qPCR confirming knockdown (KD) of *ZFAND3* in highly invasive GCSs (HI) using two ZFAND3 shRNAs ($n \geq 3$ biologically independent samples). Results are displayed as average ± SEM and were analysed with one-way ANOVA. **b, c** Decrease of ZFAND3 protein in *ZFAND3* KD by western blot ($n = 3$) and IF staining (red: ZFAND3, blue: DAPI. Scale bars = 10 μm) ($n = 2$). In analogy to patient samples, ZFAND3 protein is present in the nucleus and cytoplasm. **d** Growth curves of control and ZFAND3 KD cells show no defect in cell proliferation ($n = 3$ biologically independent experiments). **e, f** Boyden chamber invasion assay showing reduced invasion in ZFAND3 KD cells compared to control ($n = 4$ biologically independent experiments) (Scale bar = 100 μm). Results are displayed as average ± SEM and were analysed with one-way ANOVA. **g, h** ZFAND3 KD cells implanted in ex vivo brain slice cultures showed reduced colonization compared to control cells, 9 days after tumor implantation ($n = 10$ biologically independent samples) (Scale bar = 1000 μm). Results are displayed as average ± SEM and were analysed with a two-way ANOVA. **i, j** Visualization and quantification of single cell displacement of ZFAND3 KD cells in ex vivo brain slice cultures showing reduced velocity compared to control ($n = 15$ cells over 10 biologically independent samples). Results are displayed as average ± SEM and were analysed with one-way ANOVA. **k, l** Upon intracranial implantation ($n = 3$ biologically independent animals), ZFAND3 KD cell invasion toward the contralateral hemisphere was decreased compared to control tumors. GBM xenografts were analyzed at 2 months post-surgery (3 sections per mouse) (Scale bar = 100 μm). Results are displayed as average ± SEM and were analysed with an unpaired, two-sided *t* test. *$p_{value} < 0.05$, **$p_{value} < 0.01$, ***$p_{value} < 0.001$, ****$p_{value} < 0.0001$.

shCtrl HI cells (FDR < 0.05, logFC < −0.5) (Fig. 6a, b, Supplementary Fig. 7a, b). Gene ontology analysis associated these DEGs with adhesion and motility-related pathways, such as migration, integrin complex, ECM and cell adhesion (Fig. 6c). Among these DEGs, we selected genes reportedly linked to GBM cell invasion including

*COL6A2* (alpha-2 subunit of type VI collagen), *EGFR* (epidermal growth factor receptor), *FN1* (fibronectin 1), *NRCAM* (neuronal cell adhesion molecule) and *NRP1* (neuropilin 1), and confirmed their downregulation upon ZFAND3 KD (Fig. 6d–h), thus supporting a role for ZFAND3 in transcriptional regulation. To further address

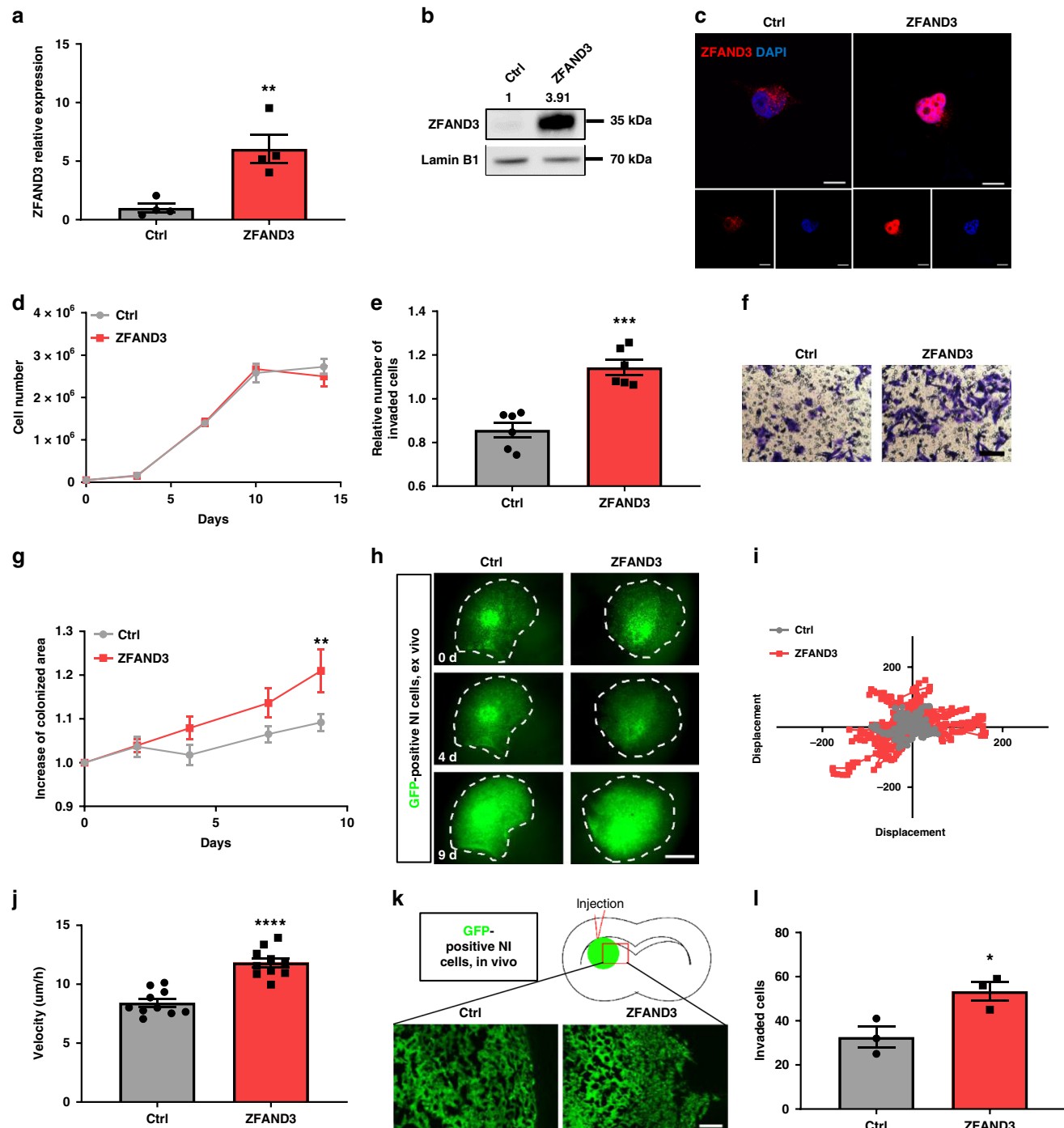

**Fig. 4 Overexpression of ZFAND3 in non-invasive GBM cells confers invasion potential.** ZFAND3 overexpression in non-invasive GSCs (NI) ($n = 4$ biologically independent samples) confirmed by qPCR (**a**), Western blot ($n = 3$) (**b**) and with IF staining (**c**) (red: ZFAND3, blue: DAPI. Scale bars = 10 μm). **d** ZFAND3-overexpression did not affect cell proliferation (n = 3 biologically independent experiments). **e, f** Boyden chamber invasion assay indicating increased invasion potential in ZFAND3-overexpressing cells ($n = 6$ biologically independent experiments) (Scale bar = 100 μm). **g, h** ZFAND3-overexpressing cells implanted in ex vivo brain slice cultures showed augmented colonization compared to control cells, 9 days after tumor implantation ($n = 10$ biologically independent samples) (Scale bar = 1000 μm). **i** Velocity of ZFAND3-overexpressing cells in ex vivo brain slice cultures was improved compared to control cells ($n = 10$, $p < 0.0001$). **j** Visualization of single cell displacement for control and ZFAND3-overexpressing cells in ex vivo brain slice cultures ($n = 15$ cells over 10 biologically independent samples). **k, l** Upon orthotopic xenografting in mice ($n = 3$), an increased number of ZFAND3-overexpressing cells invaded out of the tumor border into the surrounding tissue, compared to control cells (analysis done at 4 weeks after implantation) (3 sections per mouse) (Scale bar = 100 μm). Results of (**a, e, g, j, l**) are displayed as average ± SEM, results of (**a, e, j**, and **l**) were analysed with an unpaired, two-sided $t$ test, results of g were analysed with a two-way ANOVA. *$p_{value} < 0.05$, **$p_{value} < 0.01$, ***$p_{value} < 0.001$, ****$p_{value} < 0.0001$.

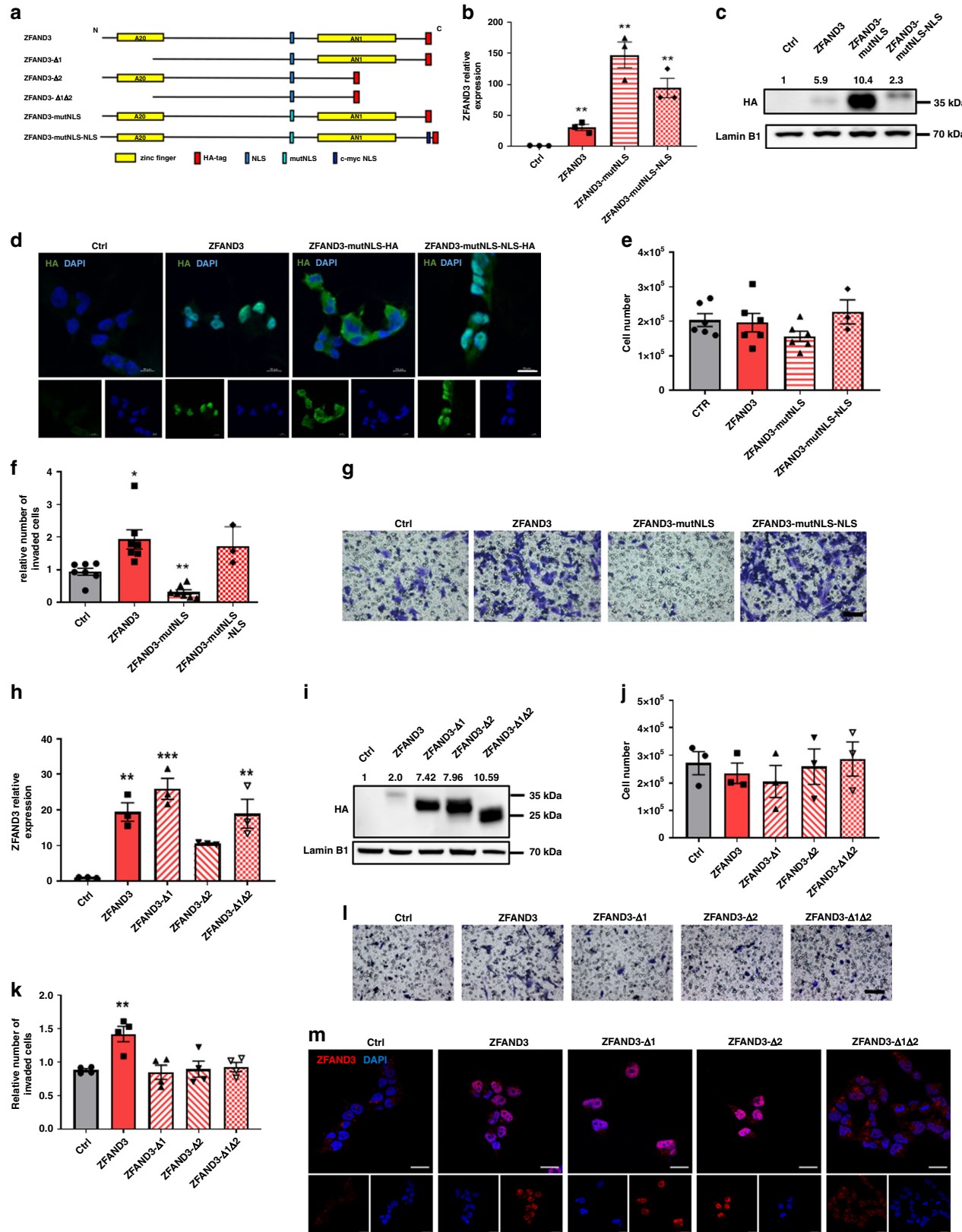

this, we interrogated the ZFAND3 neighboring interactome, using the BioID screening technology[27]. Briefly, we expressed ZFAND3 fused with a biotin ligase (ZFAND3-BirA) in GSCs, which allowed the biotinylation of proteins in close proximity (~10 nm) (Fig. 6i). After streptavidin-based purification, biotinylated proteins were analysed by mass spectrometry, and statistical analysis revealed

143 significantly enriched proteins in ZFAND3-BirA cells compared to control (adj. $p$ value < 0.05, FC > 2) (Fig. 6j). Gene ontology analysis indicated candidate interactors to be involved in RNA regulation and transcriptional processes (Fig. 6k, Supplementary Fig. 7c), reinforcing our hypothesis. To validate these results we performed ZFAND3-FLAG coimmunoprecipitation (FLAG Co-IP)

**Fig. 5 ZFAND3-induced invasion requires nuclear localization and presence of zinc-finger domains. a** Structural overview of ZFAND3 protein and corresponding mutants with zinc finger domain deletions (Δ1, Δ2, Δ1Δ2), mutated nuclear localization signal (mutNLS) and NLS rescue with c-myc NLS (mutNLS-NLS). HA-tag: hemagglutinin tag. N: N-terminus, C: C-terminus. **b** Expression of NLS constructs in NI GSCs as confirmed by qPCR ($n = 3$ biologically independent samples), results were analysed with a two-tailed, unpaired $t$ test, and (**c**) by Western blot ($n = 3$). **d** IF staining showing that ZFAND3-mutNLS does not translocate to the nucleus, while mutNLS-NLS rescue does (green: HA, blue: DAPI. Scale bars = 10 μm). ($n = 2$) **e** Proliferation assay of cells with indicated constructs control (measurements done at day 3; $n = 7$ biologically independent experiments for Ctr, ZFAND3 and ZFAND3-mutNLS, $n = 3$ biologically independent experiments for ZFAND3-mutNLS-NLS). **f**, **g** Boyden chamber invasion assay showing loss of invasion phenotype with ZFAND3-mutNLS and rescue with ZFAND3-mutNLS-NLS in NI GSC ($n = 6$ biologically independent experiments for Ctrl, ZFAND3 and ZFAND3-mutNLS, $n = 3$ biologically independent experiments for ZFAND3-mutNLS-NLS) (Scale bar = 100 μm). Results were analysed with a two-tailed, unpaired $t$ test. **h** Expression of deletion constructs in NI GSCs by qPCR ($n = 3$ biologically independent samples). Results were analysed with one-way ANOVA (**h**) Western blot (**i**) ($n = 3$). **j** Proliferation of cells overexpressing respective constructs ($n = 3$ biologically independent experiments). Results were analysed with a two-way ANOVA. **k**, **l** Boyden chamber invasion assay indicating loss of invasion induction with deletion constructs ($n = 4$ biologically independent experiments) (Scale bar = 100 μm). Results were analysed with one-way ANOVA. **m** IF staining showing that ZFAND3 Δ1 and ZFAND3 Δ2 mutants localize to the nucleus, whereas ZFAND3 Δ1Δ2 does not (red: ZFAND3, blue: DAPI. Scale bars = 20 μm) ($n = 2$) All results are displayed as average ± SEM. *$p_{value} < 0.05$, **$p_{value} < 0.01$, ***$p_{value} < 0.001$, ****$p_{value} < 0.0001$.

and subsequent mass spectrometry analysis (Supplementary Fig. 7d–f). Among the 143 proteins identified as ZFAND3 interactome, 22 proteins were also pulled-down by FLAG Co-IP supporting these as ZFAND3 interactors (Fig. 6l). These included several proteasome-associated proteins (PSMD1, PSMD8, PSMC3), nuclear importins (KPNA3, KPNA4), and splicing factors (e.g., PUF60, SF1, PPIL4) (Supplementary Fig. 7g). Since we found the invasion-related activity of ZFAND3 to be localized to the nucleus, we focused on nuclear proteins. Database-driven analysis of protein complexes (via the Dragon Database for Human Transcription Co-Factors and Transcription Factor Interacting Proteins (TcoF-DB)[28] revealed interacting proteins PUF60, Pontin and Treacle (respectively encoded by PUF60, RUVBL1 and TCOF1 genes) as common binding partners within a GPN-loop GTPase 1 (GPN1) complex. To investigate if the interaction with ZFAND3 could be further confirmed by western blot, FLAG-tagged ZFAND3 from overexpressing cells (NI-ZFAND3 OE) was immunoprecipitated and analysed. PUF60 co-immunoprecipitated with ZFAND3 thereby validating it as a ZFAND3 interaction partner (Supplementary Fig. 7h), a weak background detected in control cells was not present when using isotype-specific IgG control in overexpressing cells. Technical reasons prevented confirmation of direct interaction with TCOF1 or Pontin by WB Co-IP.

Next, we aimed to relate these findings to events taking place at the promoter regions of potential target genes. We in-silico analysed the promoters of COL6A2, NRCAM and FN1 with the Genomatix tool and identified a multitude of GC-rich target sequences, preferentially recognized by ZF containing DNA-binding proteins (Fig. 7a–c). We thus performed dual-luciferase reporter assays in U87 or HEK293T cells with the promoter sequences of COL6A2, NRCAM or FN1 and detected an increased luciferase signal upon ZFAND3 co-expression, demonstrating that ZFAND3 was able to activate the promoter region of the genes of interest and induce their transcription (Fig. 7d–f). As expected the effect was not seen with ZFAND3-Δ1Δ2 (lacking both ZF domains). We also assessed expression of COL6A2, NRCAM and FN1 genes in NI GSCs stably expressing ZFAND3 (Supplementary Fig. 8a–d) or ZFAND3 mutant constructs (Supplementary Fig. 8e–j) with no marked change in expression of target genes. Although unexpected, the latter might be related to adaptation to long term overexpression. Finally, we performed ChIP-qPCR experiments in NI GSCs overexpressing FLAG-tagged ZFAND3 or FLAG-tag alone. We observed association of ZFAND3 with in silico predicted ZF consensus sites in the promoter regions of COL6A2, NRCAM, and FN1 in ZFAND3 overexpressing cells compared to controls (Fig. 7g–i). In line, point mutations introduced into ZF consensus sites contained within respective promoter regions failed to comparably induce

luminescence in reporter assays for target genes (Fig. 7d–f). In summary, we propose that ZFAND3 induces the expression of invasion-related genes through activation of a transcriptional complex involving PUF60, ultimately boosting the invasive behavior of GBM cells (Fig. 7j).

## Discussion

ZFAND3 is a member of the ZFAND family of proteins, which contain a ZF domain of the AN1 type. ZF proteins ensure a plethora of cellular functions in health and disease, such as DNA recognition, RNA packaging, transcriptional regulation, and are involved in many aspects of cancer progression[29]. In humans, there are eight ZFAND family members, of which only ZFAND2a and ZFAND4 have recently been implicated in cancer[30–33]. Members of the ZFAND family are associated with stress response and proteasomal degradation through recruitment of the 26 S proteasome, e.g., ZFAND1[34], ZFAND2A/B[30,35], ZFAND5[36], and ZFAND6[37]. ZFAND3 has so far only been associated to type 2 diabetes susceptibility[15], but nothing is known about its cellular function or its role in tumor biology. Here, we have identified and validated ZFAND3 as a modulator of GBM cell invasion, and demonstrate that it acts through regulation of transcriptional activity. We find that in patient samples ZFAND3 expression is increased in infiltrative cells from the tumor margin.

At the mechanistic level, we show that ZFAND3-induced invasion activity relies on its nuclear localization and requires integral AN1 and A20 ZF domains. The prevention of nuclear translocation and invasive phenotype through removal of the NLS signal and rescue of localization and activity by addition of a new NLS demonstrates that ZFAND3 acts in the nucleus. This is in contrast to other ZFAND proteins that are primarily located in the cytoplasm and associated to the ubiquitin-proteasome system (UPS), which may also explain the differences observed with ZF domain mutant constructs. While single deletion of either the AN1 (ZFAND3 Δ1) or the A20 domain (ZFAND3 Δ2) did not impair nuclear translocation, the invasion phenotype was lost. This might be explained by a conformational change in the protein structure, interfering with DNA or protein binding hence affecting its activity. Functional studies of other members of the ZFAND protein family employing similar deletions have shown that proteins with deletion of one of the two domains retain certain functions. e.g., in ZFAND5 the AN1 domain is required for the stimulation of peptidase activity whereas the A20 domain is needed for the binding of polyubiquitinated proteins[36] and both ZFAND5 AN1 and A20 domains were found to be essential for RNA stabilization[38]. Also, the AN1 domain and UBL domains of ZFAND1 were found to be required for its binding to PSMD1 and p97/Cdc48, respectively[34,39]. In an attempt to further nail down

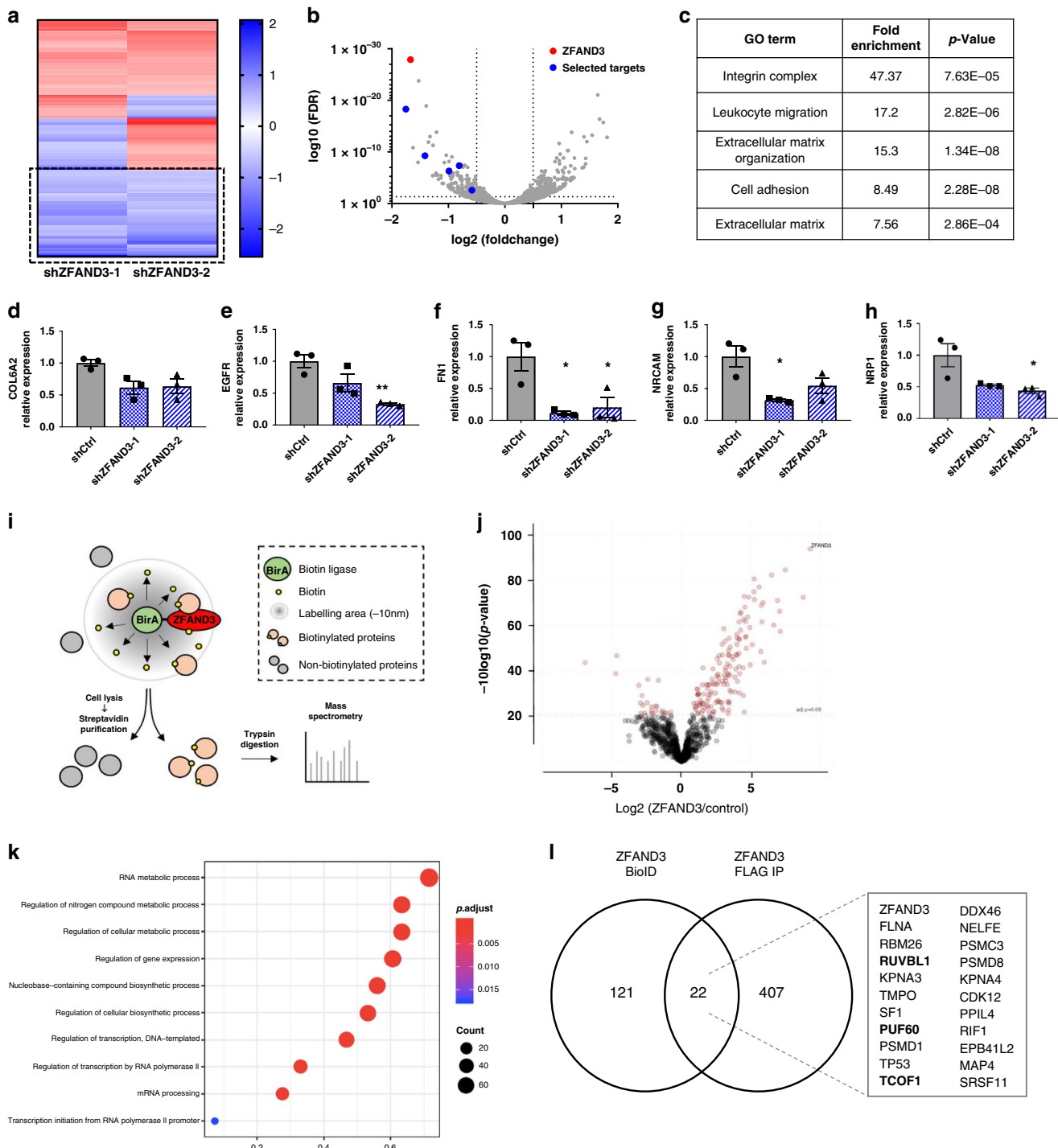

**Fig. 6 ZFAND3 regulates gene expression and is part of a nucleus-specific protein interactome involved in RNA metabolism, processing, and transcription. a** RNAseq analysis of HI GSCs (*n* = 3 per condition) identified 58 common DEGs from two KD clones (shZFAND3-1 and shZFAND3-2) vs shCtrl cells (FDR ≤ 0.05, log2FC ≤ −0.5 and ≥ 0.5). **b** Volcano plot showing DEGs in shZFAND3-2 compared to Ctrl. Among these, 5 downregulated DEGs were selected for further investigation. **c** Gene ontology analysis (David database) associated the 58 DEGs to invasion-related GO terms. qPCR confirmed reduced expression of *COL6A2* (**d**), *EGFR* (**e**), *FN1* (**f**), *NRCAM* (**g**) and *NRP1* (**h**) in HI GSCs with shZFAND3-1 or shZFAND3-2 compared to shCtrl (*n* = 3 biologically independent samples). Results are displayed as average ± SEM. Results were analysed with one-way ANOVA. **i** Bio-ID approach applied to unveil the interactome of ZFAND3 (adapted from Vernaite et al.[59]) (*n* = 4). **j** Volcano plot showing 143 proteins significantly enriched in ZFAND3-BirA expressing cells. Two-sample t test was performed with a Benjamini-Hodgberg based FDR < 0.01. **k** Gene ontology analysis demonstrated that these 143 proteins are involved in RNA metabolism, processing, and transcription. **l** Co-IP/MS in ZFAND3-FLAG expressing cells confirming 22 proteins from the Bio-ID experiment as candidate ZFAND3 interactors. Gene names of identified proteins are indicated, with Pontin (*RUVBL1*), PUF60 (*PUF60*) and Treacle (*TCOF1*) highlighted in bold. *$p_{value}$ < 0.05, **$p_{value}$ < 0.01.

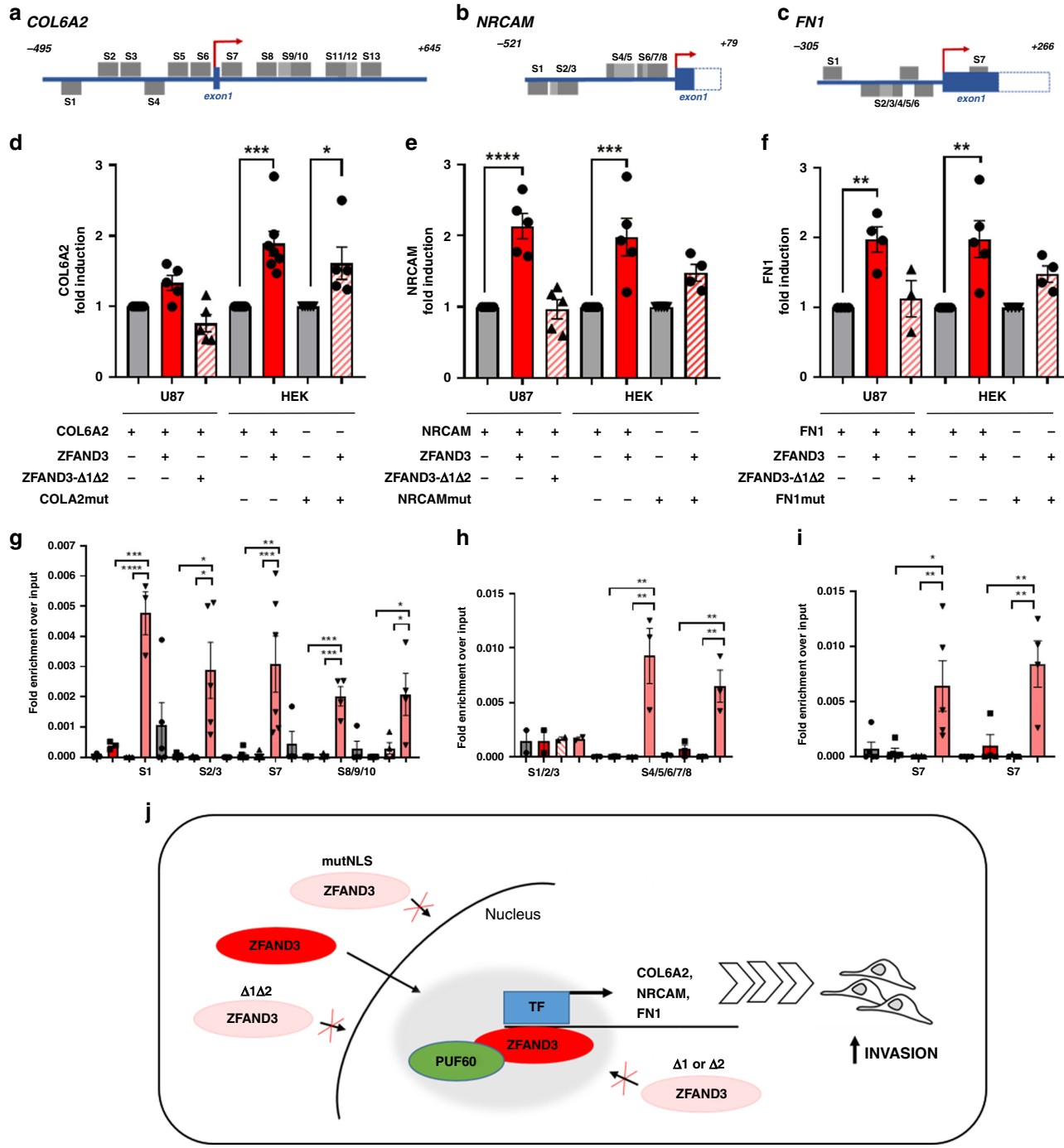

**Fig. 7 ZFAND3 binds to and induces promoter activity on COL6A2, NRCAM and FN1 genes.** Transcription factor consensus site analysis (via Genomatix) identified several putative ZF binding sites (GC-rich regions, hereafter named S1-S13) in the promoter regions of *COL6A2* (**a**), *NRCAM* (**b**) and *FN1* (**c**). Dual-luciferase reporter assays carried out in U87 or HEK293T cells showed that transiently coexpressed ZFAND3 binds to the promoter regions of *COL6A2* ($n \geq 5$) (**d**), *NRCAM* ($n \geq 4$) (**e**) and *FN1* ($n \geq 4$) (**f**) inducing luciferase expression, whereas deletion mutant ZFAND3-Δ1Δ2 did not. Luciferase activation was compromised on promoters with mutated GC-rich regions (COLA2mut, NRCAMmut, FN1mut) ($n \geq 4$ biologically independent experiments *$p_{value} < 0.05$, **$p_{value} < 0.01$, ***$p_{value} < 0.001$, ****$p_{value} < 0.0001$). ZFAND3 binding to GC-rich regions in the promoter regions of *COL6A2* ($n \geq 3$) (**g**), *NRCAM* ($n \geq 2$) (**h**) and *FN1* ($n \geq 4$) (**i**) in HI cells overexpressing ZFAND3 as determined by ChIP qPCR ($n \geq 2$ biologically independent experiments *$p_{value} < 0.05$, **$p_{value} < 0.01$, ***$p_{value} < 0.001$, ****$p_{value} < 0.0001$). **j** Proposed model of ZFAND3 activity in a nuclear protein complex that regulates a gene expression program to trigger cell invasion.

the active residues, we introduced missense mutations in the zinc-complexing amino acids of both domains (constructs M1, M2, M1-M2) similar to what was reported for ZFAND5[36]. The equivalent mutation in the A20 domain of ZFAND5 (M1)

abolished its ubiquitin-binding activity[36,40], whereas the missense mutations in the AN1 domain (M2) resulted in a loss of its ability to stimulate the proteasome[36]. In our hands ZFAND3 with either mutated domain (M1 or M2) or both (M1-M2) did not diminish

its ability to promote invasion (Supplementary Fig. 5b–f), suggesting that these specific residues are not relevant for the observed nuclear ZFAND3 activity.

In line with its nuclear activity, we identify a nucleus-specific ZFAND3 interactome, including DNA/RNA binding proteins, splicing factors and helicases generally involved in RNA metabolic processes and transcription. Based on our proteomics data and on available database-knowledge, we propose a ZFAND3 containing protein complex including PUF60 (PUF60) and possibly Pontin (RUVBL1) and Treacle (TCOF1) that regulates gene transcription (Fig. 7j). Of interest, these three proteins have been previously involved in cell motility in cancer or during development[41–43]. Binding sites for ZF containing DNA-binding proteins are enriched in the promoter regions of COL6A2, NRCAM, or FN1, which were previously described in specific invasive signatures in GBM[44,45]. We demonstrate binding of ZFAND3 to the promoter regions of these genes and confirm ZFAND3-induced transcription by dual-reporter gene assays, indicating a role for ZFAND3 in a transcriptional invasion program. In analogy to other ZFAND members, several UPS-associated proteins i.e., PSMD1, PSMC3, and PSMD8, were also identified in our ZFAND3 interactome, which may point to additional cellular functions in the cytoplasm. In this context, it has been reported that the UPS plays an essential role in transcription by maintaining a fine-tuned equilibrium of protein synthesis and degradation[46]. Whether ZFAND3's transcriptional activity is linked to a proteasomal function remains to be determined.

Collectively, these studies identify ZFAND3 as a transcriptional regulator of invasion-related genes, thereby boosting the infiltrative phenotype of GBM cells. We show that ZFAND3 requires nuclear translocation and functional ZF domains for displaying gene regulatory activity. The specific ZFAND3 interactome points to a role in transcriptional complex regulation, although an involvement in proteolysis cannot be excluded. These data expand our knowledge on the cellular functions and tumorigenic potential of ZFAND family members thus far largely known for their role in protein degradation. Further elucidation of the molecular determinants of ZFAND3 activity in oncogenic pathways may provide opportunities for targeting cancers, including GBM, prone to locoregional invasion or metastasis.

## Methods

**Cell culture.** Patient-derived GSC lines were generated in the laboratories of Christel Herold-Mende and Rolf Bjerkvig. They were characterized as non invasive (NI: NCH644; NI-2: GG6), low invasive (LI: NCH421k), and highly invasive (HI: NCH601; HI-2: NCH465; HI-3: NCH660h and HI-4: BG7) (Fig. 1 and Supplementary Fig. 1). NCH421k, NCH601, NCH465 and NCH660h were cultured as nonadherent spheres in DMEM-F12 medium (Lonza) containing 1x BIT100 (Provitro), 2 mM Ultraglutamine, 30 U/ml Pen-Strep, 1 U/ml Heparin (Sigma), 20 ng/ml bFGF (Miltenyi, 130-093-841) and 20 ng/ml EGF (Provitro, 1325950500). NCH644, BG7 and GG6 were grown in Neurobasal base medium (Life Technologies) supplemented with 1x B27 without vitamine A (Life Technologies) 2mM L-Glutamine, 30 U/ml Pen-Strep, 1 U/ml heparin, 20 ng/ml bFGF and 20 ng/ml EGF. Human embryonic kidney 293 (HEK293T) cells and U87 GBM cells were cultured in DMEM (Lonza) supplemented with 10% fetal bovine serum (FBS), 2 mM Ultraglutamine, and 30 U/ml Pen-Strep. Since GSCs were exposed to 7% FBS in the Boyden chamber assay, we verified expression of ZFAND3 and stem cell markers under these conditions (Supplementary Fig. 4f–i). No significant change in ZFAND3 expression was observed, except for HI cells (NCH601) where expression was slightly increased. Stem cell markers showed variable expression across cell lines and in response to FBS, in line with previous reports[47] (Supplementary Fig. 4f–i).

**GBM intracranial xenografts in mice.** GSCs were implanted in the brain of NOD/ SCID mice (3 mice/group) (Charles River). Mice were anesthetized with a mixture of ketamine (100 mg/kg) and xylazine (10 mg/kg) and placed in a stereotactic frame (Narishige Group, Tokyo, Japan). Tumor cells were implanted into the right frontal cortex using a Hamilton syringe (Hamilton, Reno, NV, USA) (NI, LI, NI ctrl, NI ZFAND3: 50'000 cells, HI, HI-2, HI shCtrl, HI shZFAND3: 300'000 cells). All tumor cells were GFP-labeled. Mice were sacrificed at defined time points and brains were extracted and either frozen for cryosections or embedded in paraffin. All

procedures were in accordance with national legislation and the European Directive on animal experimentation (2010/63/EU) and were approved by the responsible authorities in Luxembourg (protocol: LUPA2017/15).

**Invasion assays**

*In vitro.* For Boyden chamber assays, GSCs were plated on transwell chambers (1500 or 3000 cells/mm $^2$) (Thincert, Greiner), previously coated with a 1:1 mixture of 0.05 mg/ml collagen type I (Sigma-Aldrich) and 0.5 mg/ml protein of ECM gel (Sigma-Aldrich) in 1:1 PBS-DMEM-F12, for 2 h at 37 °C. Medium with 10% FBS was added as a chemoattractant to the lower chamber. After 3 days, cells were fixed with 4% PFA and stained with 0.05% crystal violet solution for 15 min. Non-invading cells were removed, and invasion was evaluated by counting the cells on the lower side of the membrane under light microscope and ×20 magnification (5 representative fields per membrane). Experiments were conducted in at least 3 biological replicates (each with 2 technical replicates). To account for the variability between separate assays, the data are represented as number of invaded cells relative to the global mean (total number of invading cells) of each experiment, allowing to combine biological replicates and apply appropriate statistics. The effect of Boyden chamber conditions on ZFAND3 expression and stem cell markers were verified by qPCR (Supplementary Fig. 4f–i). For the sprouting assay, $5 \times 10^3$ cells/ well were plated in an agar-coated well for 3 days to form a sphere. The sphere was transferred to an ECM/collagen-coated well, and covered with ECM. Pictures were taken using EVOS microscope, and invasion was calculated by the ratio of the sphere area at 48 h to 0 h. Analysis was corrected for the expansion of the sphere core.

*Ex vivo.* Brain slices were prepared from adult NOD/SCID mice. Briefly, mice were sacrificed by cervical dislocation, and brains were rapidly harvested and placed in ice-cold cutting solution (0.1% GlutaMax, 25 mM HEPES, 50U/ml Pen-Strep in DMEM). Brain slices (400 μm thick) were generated with a McIlwain Tissue Chopper and cultured on transwell chambers (1 μm pore size) in a 1:1 mixture of Hibernate$^{TM}$-A medium (ThermoFisher) supplemented with 20% BIT-100 and 100U/ml Pen-Strep with and DMEM-F12 supplemented with 20% BIT-100, 100U/ ml Pen-Strep, 200 mM ultraglutamine and 1U/ml heparin.

Brain slices were cultured 4 days prior to GSC implantation. For this, $5 \times 10^4$ GFP-labeled cells in 1 μl of ECM were injected in the cortex, above the corpus callosum, by gently punching a hole in the tissue and slowly releasing the cell/gel mixture. For the quantification of colonized area, pictures were taken every 2–3 days using the EVOS microscope and analysed with ImageJ. After 9 days, live cell imaging using Incucyte was performed, and the velocity of GFP positive invading cells in the brain slices were assessed with ImageJ (15 cells per condition and 10 replicates per cell line).

*In vivo.* For invasion analysis of HI and NI cells in orthotopic xenografts ($n = 3$), 8 μm cryosections were used. For each xenograft 3 sections were counted (in total 9 per condition). Sections at the level of the injection side were choosen and all GFP positive HI cells were counted manually in the contralateral hemisphere as shown in Supplementary Fig. 4j. For invasion analysis of NI cells, all GFP positive NI cells escaping the tumor mass were manually marked and counted (example shown in Supplementary Fig. 4k, l). Imaging was performed on a Ni-E microscope (Nikon) with the NIS Elements Software.

For invasion assays, data were analyzed using the GraphPad Prism 8 software. Results are reported as mean ± standard error of the mean, and n is described as the number of biological replicates. Data were analyzed with two-tailed Student *t* test. Statistical significance was set at $p < 0.05$.

**Genome-wide shRNA library screen.** A lentiviral-based genome-wide shRNA pooled library contained 95'700 shRNAs targeting 18.205 human genes (5 shRNAs/gene) (RHS6083, Dharmacon). Loss-of-function screen was performed in triplicates starting with $4 \times 10^7$ HI cells (MOI 0.3, fold representation of shRNAs 100x). After 3 days of puromycin selection (1.5 μg/ml), a reference sample (baseline control) was collected ($2 \times 10^7$ cells). Next, $5 \times 10^5$ HI cells were plated per transwell chambers (in total 24 transwell chambers per replicate to ensure 100x fold representation of shRNAs) (see protocol below) and after 3 days, invasion-deficient and invasion-competent cells were harvested, genomic DNA was extracted and shRNA sequencing libraries were prepared by PCR amplification according to the manufacturers protocol (RHS6083, Dharmacon). Libraries were pooled and 50 bp single-end sequencing was performed on an Illumina HiSeq 2500 platform (Illumina, Inc). The number of reads per sample were adapted according to the manufacturers protocol depending on the numbers of cells initially used for library preparation (per $10^6$ cells, about $10^7$ reads were sequenced). The protocol followed standard procedures and library representation rules of shRNA pooled screens[9]. The comparison of shRNA representation in invasion-deficient and invasion-competent cells was performed using the combination of four standard analysis methods: redundant siRNA activity (RSA)[19], RNAi gene enrichment ranking (RIGER)[20], Model-based Analysis of Genome-wide CRISPR-Cas9 Knockout[21,22] and HiTSelect[23]. The analysis focused on shRNAs significantly enriched in invasion-deficient cells. Baseline reference was used to verify shRNA representation after transduction. Only gene candidates with significant differential

shRNA representation in all four analysis methods were considered as hits. The total screen was performed in three independent biological replicates.

**Immunohistochemistry on GBM xenografts.** Coronal sections (8 μm) from paraffin-embedded mouse brains were pretreated for 5 min with Proteinase K (Dako) followed by 20 min incubation at 95 °C in retrieval solution (Dako). The Dako Envision + System-HRP was used following the manufacturer's instructions. Primary antibodies were incubated for 2 h at RT, and secondary antibodies for 1 h at RT (list of antibodies in Supplementary Table 3). Signal was developed with 3,3′-diaminobenzidine (DAB) chromogen in 5–10 min, and pictures were acquired using a Ni-E microscope (Nikon).

For immunofluorescent staining, cells plated on ECM coated coverslips and cryosections of orthotopic xenografts were fixed and permeabilized in TBS with 0.1% Triton (TBST), followed by blocking with 10% FBS for 1 h. Primary antibodies were incubated overnight at 4 °C (list of antibodies in Supplementary Table 3). Secondary antibodies were applied with DAPI for 1 h at RT. The cells and cryosections were then washed and mounted with Fluoromount (Sigma). Imaging was performed on a Ni-E microscope (Nikon) with the NIS Elements Software or LSM880 confocal microscope (Zeiss) with the Zeiss Zen Software. Data were analyzed using the GraphPad Prism 8 software. Results are reported as mean ± standard error of the mean, and n is described as the number of biological replicates. Data were analyzed with two-tailed Student t tests or ANOVA. Statistical significance was set at $p < 0.05$. Immunofluorescent staining for ZFAND3, IBA1 and NeuN was done as described below for clinical samples.

**Immunohistochemistry on GBM clinical samples.** Human tissue samples for immunohistochemistry were collected at Odense University Hospital, Odense, Denmark between 2010 and 2014 with approval from the Danish Data Inspection Authority (approval number 16/11065) and the Regional Scientific Ethical Committee of the Region of Southern Denmark (approval number S-20150148). All samples were diagnosed as GBM according to the 2016 WHO classification. 21 GBM with ≥60% P53-positive cells in central tumor areas and simultaneously including areas with diffuse tumor cell infiltration and tumor periphery were subject to double immunofluorescence stainings for P53 and ZFAND3. P53 protein expression was used to localize tumor cells, and thereby enable measurement of ZFAND3 protein expression in both tumor cells and nontumor cells throughout the different tumor regions. Tissue sections of 3 μm were cut on a microtome, and subject to deparaffinization and heat-induced-epitope retrieval with CC1-buffer (Ventana Medical Systems). Tissue sections were then stained with primary ZFAND3 antibody (clone: HPA016755, Atlas antibodies, 1:600 for regular immunohistochemistry, 1:1000 for immunofluorescence) for 32 minutes on the Ventana Discovery Ultra platform (Ventana Medical Systems). Antibody detection was performed with the Optiview-DAB detection system for regular immunohistochemistry and DISCOVERY OmniMap anti-Rb HRP coupled with the DISCOVERY Cy5 Kit for immunofluorescence. For double-fluorescence stainings, tissue sections were subject to a second heat-induced epitope retrieval and then incubated with either primary P53 antibody (clone: DO7, Ventana Medical Systems, ready-to-use) for 4 min, Iba1 antibody (Wako Pure Chemical Industries, 1:3000) for 16 minutes or NeuN (clone: A60, Chemicon, 1:500) for 32 min. Detection was performed with DISCOVERY OmniMap anti-Ms/anti-Rb HRP coupled with the DISCOVERY FAM Kit. Nuclei were counterstained with DISCOVERY QD DAPI and slides mounted with VECTASHIELD mounting medium without DAPI.

Fluorescence images were acquired with a Leica DM6000B microscope with an Olympus DP72 camera. Regions of interest including central tumor area, areas with intermediate tumor infiltration and peripheral tumor areas were manually outlined for each slide using the Visiopharm software V6.6.1 (Visiopharm, Hoersholm, Denmark). The software was set to randomly sample 10 images from each defined region of interest on each slide at 20X magnification. A software based cell-classifier was programmed to identify P53-positive tumor cells, DAPI-positive cells and ZFAND3-staining. Finally, the fractions and mean intensities of ZFAND3-positive tumor- and nontumor cells, including tumor cells stratified by nuclear/cytoplasmatic ZFAND3 staining were measured by the classifier (Supplementary Fig. 3a, b). Four tumors did not have ZFAND3 protein expression and were excluded from data analysis. Comparison of multiple groups was performed with ANOVA and Tukey´s post-test for data with Gaussian distribution and Kruskal-Wallis test with Dunn´s post-test for data without Gaussian distribution. Comparison of two groups was performed with T-tests for data with Gaussian distribution and Mann Whitney tests for non-Gaussian distributed data.

**ZFAND3 constructs.** ZFAND3 deletion constructs included deletion of ZF domain AN1, A20 or both. In the mutant constructs putative Zinc complexing amino acids (AAs) were mutated to alanine: C32A + C35A (M1); C176A + C181A in (M2); all four (M1-M2). For the mutant NLS constructs, the three amino acids K132, R133 and R135 were mutated to Alanine (wildtype NLS peptide: SPVKRPRLL, mutNLS peptide: SPVAAPALL). The NLS sequence PAAKRVKLDG (coding for the c-Myc NLS) was added to the C-terminus of the ZFAND3mutNLS N-terminal of the HA-tag. Mutant constructs are shown in Supplementary Fig. 5

and listed in Supplementary Table 4. shRNAs against ZFAND3 are shown in Supplementary Table 2.

All constructs were transduced in different patient-derived GSCs using lentiviral vectors. Lentiviral particles were produced in HEK cells by co-transfection of the plasmid of interest with the viral core packaging construct pCMVdeltaR8.74 and the VSV-G envelope protein vector pMD.G.2 as previously described[48]. Supernatant containing viral particles was used to transduce $4 \times 10^5$ GSCs and puromycin selection (1 μg/ml) was applied to obtain stably transduced cells.

**Quantitative real-time PCR.** 1 μg of total RNA was extracted using Qiagen RNeasy Mini Kit (Qiagen) and reverse transcribed to cDNA using the iScript cDNA Synthesis Kit (Bio-Rad) or SuperScriptIII with random primers (Invitrogen), according to the manufacturer's protocol. qPCR was carried out using Fast SYBR Green Master Mix and the Viia 7 Real Time PCR System (Life Technologies; Ta = 60 °C). qPCR reaction was performed in 5 μL volume. Fold-change (FC) was calculated using the ΔCt method and normalized to the expression of EF1α. See Supplementary Table 1 for the list of primers. CT-values were determined using the QuantStudio real time software and data were analyzed using Microsoft Excel and the GraphPad Prism 8 software. Results are reported as mean ± standard error of the mean, and n is described as the number of biological replicates. Data were analyzed with two-tailed Student t tests or ANOVA. Statistical significance was set at $p < 0.05$.

**Western blot.** Total proteins from GSCs with ZFAND3 manipulations were extracted in RIPA lysis buffer and quantified using Bradford reagent (BioRad). After SDS-PAGE electrophoresis, proteins were transferred on nitrocellulose membrane (iBlot, Life Technologies). Primary antibodies were diluted in blocking solution (milk 5%) and incubated overnight at 4 °C (list of antibodies in Supplementary Table 3). After incubation with HRP-conjugated secondary antibodies, the chemiluminescent signal was recorded using the Pierce ECL Western blot detection kit (ThermoFisher), and the ImageQuant LAS4010 imaging station and software (GE Healthcare) and Image J.

**Microarray analysis.** Genome-wide expression profiles of NI, LI, and HI were determined using the GeneChip Human Gene 1.0ST Arrays (Affymetrix), as described[49]. Briefly, total RNA was extracted using QIAGEN RNeasy Mini Kit (Qiagen), processed using the Affymetrix WT Expression kit before being hybridized on Affymetrix GeneChip Human Gene 1.0 ST arrays, according to the manufacturer's instructions (protocol P/N 702808 Rev.6). Upon hybridization, microarrays were washed, stained and scanned according to manufacturer's standard procedures. Microarray data are available in the Gene Expression Omnibus (GEO) repository under accession number GSE134470.

**RNA sequencing.** For RNA sequencing, total RNA was exctracted from shCtrl, shZFAND3-1 and shZFAND3-2 highly invasive cells using RNeasy Plus Mini Kit (Qiagen) according to manufacturer's protocol. RNA purity was assessed using the Agilent 2100 Bioanalyzer. 500 ng total RNA with a RIN of at least 9.7 were used for RNA sequencing. TruSeq library preparation on 9 samples (3 replicates per group) was performed according to the Illumina standard protocol. Paired end of 2x 75 bp reads was performed using the NextSeq500 (Illumina). Raw sequencing data was mapped to human genome (assembly version Genome Reference Consortium Human build 38, GRCh38) using Bowtie v2-2.3.2[50]. Transcript abundance was evaluated with HTSeq v0.6.1[51]. The normalization and transcripts differential expression analyses of count data were performed using the DESeq2 package[52]. Comparisons of shZFAND3-1 vs. shCtrl and shZFAND3-2 vs. shCtrl were assessed with the following parameters: fold change > log2 −0,5 fold, false discovery rate 0.05. Common differentially expressed genes in both comparisons were considered as hits (Supplementary Table 6). RNAseq data are available in the GEO repository under accession number GSE138618.

**Streptavidin enrichment of biotinylated proteins and on-bead digestion (BioID).** For the analysis of ZFAND3 interactome, NI GBM GSCs (NCH644) were stably transduced with the myc-BioID-ZFAND3 construct (Supplementary Table 4), nontransduced cells were used as control. To activate the biotin ligase, cells were incubated in complete media supplemented with 100 μM D-Biotin (Carl Roth) for 48 h. After two PBS washes, cell pellets were lysed in 1% sodium deoxycholate in 50 mM ammonium bicarbonate pH8 and protease inhibitor cocktail, sonicated, incubated 30 min on ice and centrifuged at 11,000 × g and 4 °C for 30 min. Supernatant was recovered and proteins were quantified using PierceTM BCA Protein Assay Kit (Thermo Scientific). PierceTM Streptavidin Magnetic Beads (Thermo Scientific) were washed once in cold lysis buffer, and 1.8 mg of protein per sample was incubated with 50 μL of beads for 3 h at 4 °C with gentle end-over-end rotation. Beads were captured with a magnetic stand, washed 3 times in cold lysis buffer and 3 times in cold 50 mM ammonium bicarbonate pH8. Proteins were digested overnight on beads at 37 °C and 60 xg with 1 μg of trypsin (Promega). Fresh trypsin (0.5 μg) was added to the beads and incubation was extended for 2 h at 37 °C and 60 xg. Beads were captured with a magnetic stand and peptides were transferred to a new tube. Beads were rinsed twice with 100 μL

of 50 mM ammonium bicarbonate and these 2 rinses were pooled to the previously eluted peptides. Samples were further reduced with 10 mM DTT for 45 min at 37 °C and 60 xg, and carbamidomethylated with 25 mM iodoacetamide for 30 min at RT and 60 xg in the dark. Peptides were finally centrifuged 15 min at 11,000 × g at 4 °C to remove any insoluble material.

**ZFAND3-FLAG Coimmunoprecipitation for mass spectrometry**. Co-Immunoprecipitation (Co-IP) for mass spectrometry (MS) was done on whole cell lysate using the Dynabeads™ Co-IP Kit (Invitrogen, 14321D). For antibody coupling, 1.5 mg Dynabeads™ M-270 Epoxy were incubated with 8 μg of FLAG M2 antibody (Sigma F1804) at 37 °C on a mixing roller for 16 h. Beads were washed according to manufacturer's instructions and diluted to a concentration of 10 mg/ml and left at 4 °C until further processing. NI GBM GSCs overexpressing ZFAND3 with a FLAG-tag were incubated in complete media supplemented and while harvesting washed twice in ice-cold PBS. Cells were lysed with 1x IP Buffer containing NaCl 100 mM, DTT 1 mM and one cOmplete™ EDTA-free Protease Inhibitor Cocktail for 15 min on ice, and sonicated. The lysates were quantified using the Bradford Reagent (Bio-Rad). Proteins of interest were captured by incubating 1 mg of protein lysate with 1.5 mg of antibody-coupled beads at 4 °C for 30 min. Beads were resuspended in cold ammonium bicarbonate 50 mM, reduced with DTT 10 mM for 45 min at 37 °C and 60 xg, and carbamidomethylated with iodoacetamide 25 mM for 30 min at RT and 60 xg in the dark. Proteins were digested with 1 μg of trypsin (V5111, Promega) overnight on beads at 37 °C and 60 xg. Fresh trypsin (0.5 μg) was added to the beads and incubation was extended for 2 h at 37 °C and 60 xg. Dynabeads were captured with a magnetic stand, peptides were recovered and transferred to a new tube. Beads were rinced twice with 100 μL of ammonium bicarbonate 50 mM and these 2 rinses were pooled to the previous eluate. Peptides were centrifuged 15 min at 11,000 × g at 4 °C to remove any insoluble material, acidified in 1% formic acid, desalted on Sep-Pak tC18 Elution Plates (Waters, 186002318), dried by vacuum centrifugation and reconstituted in 12 μL of 1% Acetonitrile/0.05% trifluoroacetic acid. Samples were quantified by Nanodrop and further analysed by mass spectrometry (see below).

**Mass spectrometry analysis and data analysis**. Peptides were acidified in 1% formic acid, desalted on Sep-Pak tC18 μElution Plates (Waters, 186002318), dried by vacuum centrifugation and reconstituted in 20 μL of 1% Acetonitrile / 0.1% formic acid. 1 μL of each sample was measured by LC-MS/MS on a Q-Exactive Plus mass spectrometer (Thermo) connected to a Dionex Ultimate 3000 (Thermo), run in trap mode using a Acclaim Pepmap 100 trap column (Dionex). The peptides were separated on a 15 cm Acclaim pepmap RSLC column (Dionex) using a 89 min gradient (2% to 50% acetonitrile) with a flow rate of 0.3 μl/min. MS acquisition was performed with an MS1 resolution of 70,000 and a scan range from 375 to 1500 m/z with an AGC target of $3 \times 10^6$. The top 12 peaks were selected for fragmentation in data-dependent mode using an MS2 resolution of 17,500 and a maximum injection time of 45 ms and an isolation window of 1.2 m/z and an AGC target of $10^5$. Dynamic exclusion was set to 20 s and the normalized collision energy was specified to 28.

For analysis, the MaxQuant software package version 1.6.5.0[53] was used. Carbamidomethylation on cysteine was set as a fixed modification and oxidized methionine and acetylated N-termini as variable modifications. Peptide tolerance was 20 ppm and the minimum ratio for LFQ was set to 2. An FDR <1% was applied for peptides and proteins and the Andromeda search[54] was performed using a human Uniprot database (July 2018). MS intensities were normalized by the MaxLFQ algorithm implemented in MaxQuant[55] while using the match-between-runs feature.

Only protein groups that were not marked as potential contaminants nor detected by MaxQuant using reverse database were kept for the analysis. LFQ-normalized intensities were log2-transformed. Statistical analysis of the dataset was performed using R-statistical software package (version 3.4.1). For data analysis, first, proteins that were only identified by site or were potential contaminants were excluded. Only the proteins that were found in at least three biological replicates for every condition were used for column wise imputation from a normal distribution and subsequent statistical analysis. Significantly differentially expressed protein groups were detected by R-package limma[56]. For this, a two-sample *t* test was performed with a Benjamini-Hodgberg based FDR <0.01. Abundance changes with a *p* value < 0.05 and a minimum fold change of 2 were considered significant (Supplementary Table 7 for BioID experiment, Supplementary Table 8 for IP experiment). The GO analysis was carried out by R software, *p* value and *q* value cutoff 5%, n and minimum 3 proteins per category as threshold: significantly, differentially regulated proteins (corresponding Uniprot IDs) were used as targets and all other proteins captured by proteomics experiment were considered as a background. Protein IDs that matched several gene IDs were removed from the analysis.

**ZFAND3-FLAG coimmunoprecipitation for western blot analysis**. Co-IP with subsequent Western blot analysis was performed on nuclear extracts following Agoston and Schulte[57]. For antibody coupling, 1 mg Dynabeads™ M-270 Epoxy were incubated with 5 μg of FLAG M2 antibody (Sigma F1804) or Mouse IgG at 37 °C on a mixing roller for 16 h. Fresh nuclear proteins extracts from NI GBM

GSCs overexpressing ZFAND3 with a FLAG-tag and Ctrl-FLAG expressing cells were prepared as following. $2 \times 10^6$ were used to prepare cytoplasmic lysates in 300 μl Buffer A + (10 mM HEPES, 10 mM KCl, 1 mM DTT, 1x cOmplete™ EDTA free), followed by 15 min incubation on ice 30 μl IGEPAL/PBS solution was added and cells were briefly vortexed. The nuclei were pelleted by centrifugation for 1 min at 9000 × g at 4 °C. Nuclear extract lysis was done by adding 180 μl Buffer B + (10 mM HEPES, 10 mM KCL, 1 mM DTT, 400 mM NaCl, 1% IGEPAL, 1x cOmplete™ EDTA free) and incubated for 15 min on a rotating wheel. To obtain isotonic salt concentration 300 μl Buffer A + was added. Lysates were treated with DNase I (Ambion) with 50U for 30 min at 4 °C. Nuclear protein extracts were pre-cleared for 2 h using 0,3 mg Dynabeads™ M-270 Epoxy at 4 °C on a rotating wheel. 30 μl of each lysate was kept as input control for later. Dynabeads were equilibrated in 900 μl Buffer BP + (3:1.8 ratio of Buffer A and B) before adding the lysates. Nuclear protein extracts were divided in two (FLAG and IgG) to use equal amount of lysate per IP. Dynabeads-antibody containing lysates were incubytes for 2 h at 4 °C on a rotating wheel. After magentic sepatation of precipitate and supernatant, supernatant was removed and kept for later as supernatant control. Dynabeads-bound precipitates were washed four times with Buffer BP + and once with 1x LBW 0,1 % Tween-20. After the washing steps, 30 μl 4x LDS (50 mM DTT) was added to each sample and incubated for 10 min at 70 °C. After magnetic separation, supernatants were transferred into a fresh tube. Samples were directly loaded on a SDS-PAGE for Western Blot analysis.

**Dual luciferase reporter assay**. Plasmids encoding the Firefly Luciferase reporter gene under the control of the candidate gene promoters, and a Renilla Luciferase plasmid under the control of an SV-40 Promoter were purchased from Vector-Builder®. Plasmid DNA was purified using NucleoBond Xtra Midi Kit (Macherey-Nagel). Reporter cells (U87 or HEK293T) were seeded into 24-well plates at a density of $7.5 \times 10^4$ per well and transfected the next day, using 1.5 μl Lipofecta-mine 2000 (Thermo Fisher), with the selected Luciferase reporter plasmids (COL6A2, FN1 and NRCAM) and co-transfected with either empty control, ZFAND3 or ZFAND3 Δ1Δ2. A Renilla Luciferase Reporter was co-transfected with a 1:10 ratio of luciferase reporter plasmids as a reference control. Luciferase activity was recorded 30 h post-transfection using Dual-Glo® Luciferase Assay System (Promega) according to the manufacturer's instructions, using ClarioStar plate reader (BMG LabTech). The ratio of the Firefly:Renilla luminescence was calculated and normalized to the ratio of the empty control. Data were analyzed using the GraphPad Prism 8 software. Results are reported as mean ± standard error of the mean, and n is described as the number of biological replicates. Data were analyzed by ANOVA. Statistical significance was set at *p* < 0.05.

**Chromatin immunoprecipitation (ChIP) qPCR**. ChIP was performed as described previously[58] with the following changes: cells were cross-linked for 30 min at 4 °C in 2% PFA made from freshly prepared 18.5% PFA. Chromatin was sheared to a mean length of 100–500 bp with a Bioruptor UCD-200 in a cold room at 4 °C (Diagenode). Immunoprecipitation was performed with the antibodies listed in S3. ChIP precipitates were assessed by quantitative real-time PCR with the primers listed in Supplementary Table 1 and PowerUp™ SYBR™ Green master mix reagents (Thermo Fisher Scientific) on a QuantStudio™ 5 Real-Time PCR detection system (Thermo Fisher Scientific). Enrichment of the precipitated DNA was determined relative to the input (1:100) as $100 \times 2^{(Ct\ adjusted\ input\ -\ Ct\ immunoprecipitate)}$. Results are reported as mean ± standard error of the mean, and n is described as the number of biological replicates. Statistical significance was determined using ANOVA in the GraphPad Prism 8 software.

**Reporting summary**. Further information on research design is available in the Nature Research Reporting Summary linked to this article.

## Data availability

All data generated or analysed during this study are included in this published article (and its Supplementary Data files). RNAseq data are available in the GEO repository under accession number GSE138618; DNA microarray data under accession number GSE134470.

Raw proteomics data have been deposited in the The MassIVE site repository of the University of California San Diego (ProteomeXchange consortia) under ftp://massive.ucsd.edu/MSV000086247/. The constructs generated during the current study are available from the corresponding author on reasonable request.

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

## Acknowledgements

We thank Virginie Baus and Vanessa Barthelemy for technical assistance and are grateful for the financial support of the Fondation Cancer Luxembourg (INVGBM and Pan-RTK Targeting), Télévie-FNRS (GBModImm no 7.8513.18 and TETHER no 7.4615.18) and the Luxembourg National Research Fund (FNR; CORE Junior C17/BM/11664971/DEMICS).

## Author contributions

A.S., V.N., E.K., A.M.K., A.O., M.D., C.F., A.C.H., A.G., D.P.H., S.R., B.K. performed experiments and analyzed data. P.V.N., A.M., D.P.H. performed statistical and bioinformatic analyses. C.H.M., R.B., B.W.K. provided material. G.D., B.W.K., S.P.N. supervised experiments. S.P.N., A.S. designed project. A.S., V.N., E.K., S.P.N. wrote manuscript. S.P.N. supervised project and provided funds. All authors edited and approved the manuscript.

## Competing interests

The authors declare no competing interests.
