## [Peer Review File · Nature Communications]

Reviewers' Comments:

Reviewer #1:

Remarks to the Author:

This manuscript takes a molecular genetic approach to gain a better understanding of glioblastoma invasion. Using Transwell assays the authors derive subpopulations of glioma cells in vitro that have high or low invasive potential. These were sourced from actual patient samples. They then use a RNA interference screen to identify single genes which, when deleted, retard the ability of highly invasive gliomas to do so. This led to 17 candidates one of which was novel, AN1-Type Zinc Finger protein 3 (ZFAND3). Through a series of detailed and elegant studies the authors show that this transcription factor is necessary and sufficient to make glioblastomas more invasive. They show that nuclear localization is required and that this transcription factor regulates a host of genes that play known and suspected roles in invasion.

This study adds significantly to molecular understanding of glioma invasiveness and uncovered a key driver. The study was conducted well and the conclusions are well supported by the results. The only caveat is that I do not see a plausible way to utilize this information clinically or pharmacologically. This notwithstanding, this is a well executed study that adds to our understanding of just how complex this disease is.

Reviewer #2:

Remarks to the Author:

Schuster and colleagues report on the newly discovered involvement of a zinc finger protein (ZFAND3) in the invasion of glioblastoma. To do so, they have performed a large-scale high-throughput loss-of-function screen and identified several shRNAs that impair the invasion of glioblastoma stem-like cells in vitro. The paper is generally well written, although editing by a native speaker would be of benefit. However, the manuscript has several major flaws which should be addressed.

1. Glioblastoma defined as a very heterogeneous disease, being subclassified into different sub-types based on gene expression: classical, proneural or mesenchymal. These sub-types also behave differently on a phenotypic level, including proliferative and invasive capacity. This heterogeneity is also reflected at the single cell level within a tumour. It would be interesting to define the sub-types of the LI and HI cell lines and investigate whether ZFAND3 expression correlates with the subgrouping.
2. The authors used serum as chemoattractant. However, it is well known that serum induces differentiation of GSCs. It would be important to control what happens to ZFAND3 expression with and without serum.
3. The line HI-2 is much less invasive in vitro as compared to its phenotype in vivo. Could this be an effect of cell culture with the cell line losing its invasive capacity in vitro? The authors should comment on this. The verification of the data in an additional HI GSC line would be of benefit to ensure the robustness of the findings. In addition, the luciferase assay depicted in Figure 7 was performed in U87 cells. This should be clearly stated in the text and/or in the figure legend. What was the reason for using U87? How is ZFAND3 expressed in this cell line? What is the invasive potential of U87 as compared to GSCs HI/LI?
4. In Figure S2, the y-axes seem to be inverted or at least don't match the legend ($\log_{10} > 0$). It should be corrected.

5. In all the graphs assessing the invasive potential of the cells with or without treatment, the authors present "relative number of cells." Relative to what? It is not clear how data were processed.

6. Figure 3k and Figure 4k. In order to be able to assess the importance of ZFAND3 knockdown or overexpression *in vivo*, it would be important to also have access to the staining of the complete section.

7. Figure 5a is wrongly oriented. The constructs should be displayed from the N to C terminus (N on the left; C on the right). It must be corrected.

8. It should be clearly stated which amino acids were mutated (ZFAND3-mutNLS).

9. The authors have deleted a large portion of ZFAND3 and assumed that those domains are directly involved in the function of this Zinc Finger protein. However, we cannot be sure that the phenotype observed is not simply due to the fact that those deletions dramatically impair the folding of the protein, rendering it functionally inactive.

10. The authors have hypothesized that ZFAND3 acts as a transcription factor within a complex (Figure 7). However, this finding is based on *in silico* data that were not all verified experimentally. To ensure the robustness of the data, it would be necessary to perform additional verification. The promoter sequence of COL6A2, NRCAM and FN1 harbor more than just the ZBED binding site. To ensure that the function of ZFAND3 as a transcription factor really depends on ZBED sequences, these should be mutated and no luciferase signal should be observed. Overexpression of ZFAND3 and simultaneous knockout of ZBED (1-4) should not increase the luciferase signal if the hypothesis is true.

ChIPSeq could be performed to ensure that ZFAND3 is indeed found at the promoter of the above-mentioned genes.

The authors identified/suggested several binding partners of ZFAND3. These results must be verified by co-immunoprecipitation.

Reviewer #3:

Remarks to the Author:

In this report, Schuster et al. identify ZFAND3 as a candidate driver of tumor invasion in glioblastoma (GBM) using an RNA interference screen, and then perform in-depth analysis using complementary *in vitro*, *ex vivo*, and *in vivo* approaches to validate the invasive capacity of this novel protein in patient-derived glioblastoma stem cell-like models. By manipulating ZFAND3's zinc-finger domains and nuclear localization sequence and using reporter assays, the authors go further to show that this protein is active in the nucleus and that it regulates the expression of three invasion-related genes – COL6, FN1, and NRCAM.

The drivers of tumor invasion are still poorly understood in glioblastoma, and harnessing new targets for anti-invasive therapy is an important goal in the field. The study addresses this important question in an elegant way by using an unbiased screen in a relevant model for tumor invasion to identify a target, which is then thoroughly validated by several complementary approaches in patient-derived cells.

The text is overall well written and representative of the literature in the field (with some suggestions on recent invasion literature provided below) and the experiments and conclusions are appropriately interpreted.

There are several important strengths to this study. (1) The separation into invasion-competent and invasion-defective groups provides a sound model to screen for invasion drivers. (2) The selection criterion of the 17 invasion-essential candidates is rigorous, using four different computational methods. (3) The complementary *in vivo*, *ex vivo*, and *in vitro* tools used recapitulate important aspects of GBM tumor cell invasion biology, and at least the *in vitro* analysis also accounts for aspects of tumor cell proliferation. (4) Figure 3 provides strong evidence that knockdown of ZFAND3 in highly invasive (HI) GBM cells impairs tumor migration *in vitro*, *ex vivo*, and *in vivo*. As a complement, Figure 4 also shows that overexpression of ZFAND3 in non-invasive (NI) GBM cells confers invasive potential. Importantly, the author shows that there is no effect on cell proliferation (*in vitro*). Furthermore, Figure 5 provides intriguing insight into the need for both zinc finger domains and the nuclear localization signal for pro-invasive ZFAND3 activity. A possible caveat in the results is the fact that single deletion of the zinc finger domains did not abolish nuclear translocation, yet it showed an anti-invasive phenotype, but the authors appropriately discussed the possibility for conformational change causing loss of DNA and/or protein binding as a possibility for the observed downstream effect on transcriptional activity and invasion.

My main concern is in regards to the relevance of the immunofluorescence ZFAND3 analysis in primary human samples presented in Figures 2 and S3 and its relationship to the overall message of the paper. Additional more minor concerns are outlined as well. I believe these concerns can be addressed without major revisions to this manuscript. My specific comments are provided below.

Major Concern:

1. Figure 2f-h and Supplementary Figure 3c. The immunofluorescence analysis of ZFAND3 in patient sample tissue is confusing and it does not provide very strong support for the importance of this molecule as a nuclear (transcriptional) regulator of tumor invasion. First, only rare tumor cells appear to stain positive for ZFAND3 in 2f and S3c (most P53+ tumor cells are ZFAND3-) and there are arguably more ZFAND3+P53- than ZFAND3+P53+ cells in the classifier). Second, the staining pattern of ZFAND3+ cells appears largely cytoplasmic rather than nuclear, in contrast to what is seen by later immunocytochemistry staining in NI GBM cell lines. This raises several questions that need to be reconciled in the paper.

1) What is the relevance of ZFAND3 expression in human samples if less than 1% of tumor cells show convincing protein expression? Perhaps analysis of ZFAND3 staining in a larger number of GBM samples would be more informative, such as within a tissue microarray that includes both the core and the infiltrative tumor components. 2) ZFAND3 immunostaining appears predominantly cytoplasmic – how do the authors reconcile that with ZFAND3 activity? Perhaps analysis of nuclear vs. cytoplasmic expression of ZFAND3 in GBM patient tissue samples, in the central vs. intermediate vs. periphery zones, could be a better predictor of activity than the current analysis of ZFAND3 overall intensity in Figure 2h? 3) Are ZFAND3+P53- cells considered non-tumoral? Is ZFAND3 expressed in a subset of normal cells in both human and xenograft models (see also minor comment 4)?

Minor Concerns:

1. Introduction. Line 71: consider including the recent CRISPR-Cas9 genetic screen study that also aims to discover new drivers of glioblastoma invasion (PMID: 31570734); and Line 68: consider mentioning recent studies exposing other transcriptional and cytoskeletal regulators (and potential therapeutic targets) of tumor migration/invasion in glioblastoma (PMID: 28122245, 30275445, 31235578, others).

2. Figure 1a: The endpoints for NI, LI, and HI seem to be quite different. The endpoint for NI in this figure is 5 weeks, while in later experiments (Figure 4k-l for example), the xenografts generated from

NI cells had an endpoint of 8 weeks. The much shorter endpoint for NI vs. HI may be skewing the observation that NI tumors are non-infiltrative; a later endpoint for NI should be considered.

3. Supplementary Figure 2b-d. The screen appears to only show two candidates differentially enriched in HI cells: CSF, regarded as positive control, and ZFAND3. While this does not downplay the findings in this report, some mention in the results and / or discussion in regards to this is warranted.

4. Figure 2d. In the NI and LI xenografts, some of the ZFAND3+ cells appear to show neuronal and/or microglial morphology while in the HI tissues, they appear more tumoral. It could be helpful to perform immunofluorescence co-labeling with a tumor-specific marker (human Vimentin, human nuclear antigen) and neuronal or microglial markers to define better the tumor-specific contribution of ZFAND3+ staining in these models.

4. Figure 4k-l. This is a very important finding in the paper and it would be helpful to visualize how the cells were scored by providing an actual annotated image of cells counted at the infiltrative margin / intermediate zone.

5. Figures 3+4. The authors make a point to show that ZFAND3 does not affect cell proliferation in vitro, but there are no experiments assessing the effect of ZFAND3 knockdown on cell proliferation in vivo, which could be easily accomplished by immunostaining for an accepted cell proliferation marker in the xenograft histological sections.

6. Sample number (n) is missing from several figure legends and should be added for completion. For example, Figure 6a-b legend does not indicate how many samples were analyzed for the differential expression RNA-seq analysis, although in the methods section it was indicated that samples were sequenced in triplicates.

7. Figure S3. The authors may want to also check ZFAND3 expression in the IVY GAP database, which is annotated by core/infiltrative anatomical regions.

8. Line 107: Essential misspelled on line 107.

9. Line 171: Should be CDH2 instead of CDH1 (CDH1 is E-cadherin).

RESPONSE to REVIEWERS' COMMENTS – *Schuster et al.*

Reviewer #1 (Remarks to the Author):

This manuscript takes a molecular genetic approach to gain a better understanding of glioblastoma invasion. Using Transwell assays the authors derive subpopulations of glioma cells in vitro that have high or low invasive potential. These were sourced from actual patient samples. They then use a RNA interference screen to identify single genes which, when deleted, retard the ability of highly invasive gliomas to do so. This led to 17 candidates one of which was novel, AN1-Type Zinc Finger protein 3 (ZFAND3). Through a series of detailed and elegant studies the authors show that this transcription factor is necessary and sufficient to make glioblastomas more invasive. They show that nuclear localization is required and that this transcription factor regulates a host of genes that play known and suspected roles in invasion. This study adds significantly to molecular understanding of glioma invasiveness and uncovered a key driver. The study was conducted well and the conclusions are well supported by the results. The only caveat is that I do not see a plausible way to utilize this information clinically or pharmacologically. This notwithstanding, this is a well executed study that adds to our understanding of just how complex this disease is.

Response:

We thank the Reviewer for the positive and encouraging review. We agree that the steps towards pharmacological interference are not straightforward. Nevertheless we believe that gaining insight into the mechanisms of glioma invasion and the molecular players involved will contribute to the design of future treatment strategies.

Reviewer #2 (Remarks to the Author):

Schuster and colleagues report on the newly discovered involvement of a zinc finger protein (ZFAND3) in the invasion of glioblastoma. To do so, they have performed a large-scale high-throughput loss-of-function screen and identified several shRNAs that impair the invasion of glioblastoma stem-like cells in vitro. The paper is generally well written, although editing by a native speaker would be of benefit. However, the manuscript has several major flaws which should be addressed.

1. Glioblastoma defined as a very heterogeneous disease, being subclassified into different subtypes based on gene expression: classical, proneural or mesenchymal. These subtypes also behave differently on a phenotypic level, including proliferative and invasive capacity. This heterogeneity is also reflected at the single cell level within a tumour. It would be interesting to define the subtypes of the LI and HI cell lines and investigate whether ZFAND3 expression correlates with the subgrouping.

We thank the reviewer for this interesting suggestion. Based on our available gene expression/methylation data of the cell lines (data available for 6 out of the 7 cell lines), we determined their transcriptomic subtype using the Wang et al. 2017 classification. We did not observe a consistent correlation between molecular subtype and invasion behavior. This observation has been added in Fig. S1i and in the text (page 4).

Since the relatively small sample size does not allow to draw a firm conclusion on the correlation of ZFAND3 expression and subgrouping, we investigated publicly available datasets from TCGA where we did not find a correlation of ZFAND3 expression with subtyping. We have included these data in Fig. S2f and on page 5 in the text. It should be noted however that molecular subtype determination is based on bulk expression analysis and may not reflect single cell behavior.

2. The authors used serum as chemoattractant. However, it is well known that serum induces differentiation of GSCs. It would be important to control what happens to ZFAND3 expression with and without serum.

To address this point, we analysed ZFAND3 expression by qPCR in cells exposed to the same conditions as during the invasion assay (7% FBS for 72 hours). In parallel we also assessed the expression of several stem cell markers to assess the differentiation process. ZFAND3 was slightly increased upon FBS treatment in one of the HI cells, while its expression was not changed in the other cell lines. Stem cell markers showed heterogeneous expression also in response to FBS, which is in line with our previous data (Dirkse, Golebiewska et al. 2019). These data have been included in Suppl. Fig. S4f-i (text page 12).

3. The line HI-2 is much less invasive *in vitro* as compared to its phenotype *in vivo*. Could this be an effect of cell culture with the cell line losing its invasive capacity *in vitro*? The authors should comment on this. The verification of the data in an additional HI GSC line would be of benefit to ensure the robustness of the findings. In addition, the luciferase assay depicted in Figure 7 was performed in U87 cells. This should be clearly stated in the text and/or in the figure legend. What was the reason for using U87? How is ZFAND3 expressed in this cell line? What is the invasive potential of U87 as compared to GSCs HI/LI?

We thank the reviewer for highlighting this point. Based on our data we have no indication that GSCs may lose their invasive capacity *in vitro*. As shown in Fig. 1 the invasive potential of GSCs *in vitro* and *in vivo* is generally well correlated, albeit with some variation between cell lines, which may reflect phenotypic heterogeneity of GBM. We have now added 3 additional cell lines to strengthen this point: two highly invasive cell lines (HI-3, HI-4) and one non-invasive cell line (NI-2) (shown on Fig. S1a-b). In analogy to cells on Fig. 1, NI-2 is non-invasive both *in vitro* and *in vivo*, while cell lines HI-3 and HI-4 are highly invasive *in vitro* and *in vivo*. NI is shown for comparison on Fig. S1a.

For a more comprehensive overview, all the cell lines are depicted together in the graph below, which clearly shows that HI-2 is part of the highly invasive group:

Interestingly, in contrast to GSCs, U87 cells do not seem to follow this rule. While these cells are highly invasive *in vitro*, they are well known for their circumscribed growth *in vivo*. We have previously used U87 cells in Boyden chamber assays where they showed a similar invasive behavior as HI GSCs (Meiser, Schuster et al. 2018). Because of this discrepancy between *in vitro* and *in vivo* behavior (among other reasons), we do not consider U87 cells a good model for GBM invasion studies.

In fact we only used U87 cells in Fig. 7 as a cellular readout for the luciferase assay. The reason for this being that GSCs are very difficult to transfect and therefore inappropriate for the luciferase assay. Since this assay requires a triple transfection (promoter-luciferase construct, Renilla construct and Effector construct), this was technically impossible in GSCs and we had to turn to cells that can be more easily manipulated. In these experiments, the role of the cell line is simply to provide the transcriptional machinery to show induction of promoter activity in response to ZFAND3 expression. For our newly added luciferase promoter assays, we have used HEK cells instead. This is now clearly described in the method section (page 21) and in the legend of Figure 7.

4. In Figure S2, the y-axes seem to be inverted or at least don't match the legend ($\text{Log}_{10} > 0$). It should be corrected.

We thank the reviewer for highlighting this and apologize for the inadvertency. The legend should read ($\text{log}_{10}\text{FC} < 0$) and has been corrected accordingly.

5. In all the graphs assessing the invasive potential of the cells with or without treatment, the authors present "relative number of cells." Relative to what? It is not clear how data were processed.

Each invasion experiment was conducted in at least 3 biological replicates (n=3). For each replicate, two technical replicates were prepared, and invaded cells were evaluated on 10 representative pictures (5 pictures per chamber). In order to take into account the variability between separate assays, within each experiment (biological replicate) the number of invading cells was normalized against the total number of invading cells of that experiment (global mean of the experiment). The data are represented as number of invaded cells relative to the global mean. This allowed to combine biological replicates and apply appropriate statistics. This has now been explained in the method section (page 12).

6. Figure 3k and Figure 4k. In order to be able to assess the importance of ZFAND3 knockdown or overexpression in vivo, it would be important to also have access to the staining of the complete section.

Overview images of the complete sections have now been integrated for ZFAND3 KD xenografts as well as ZFAND3 overexpression xenografts (Fig. S4j-k). (see also response to Reviewer 3 below).

7. Figure 5a is wrongly oriented. The constructs should be displayed from the N to C terminus (N on the left; C on the right). It must be corrected.

We apologize for this confusion. The constructs on Figure 5a were correctly displayed from N to C, but there was a mistake in the description in the text (A20 is N-terminal, while AN1 is C-terminal). This has now been corrected. Furthermore we have updated the figure to include the newly generated constructs and have clarified the orientation N-C in Fig.5a.

8. It should be clearly stated which amino acids were mutated (ZFAND3-mutNLS).

Amino acid mutations of all ZFAND3 constructs including the mutant NLS construct and additional newly generated constructs have been added to the method section (page 16). For a better overview, we have also prepared a figure indicating all mutations at the amino acid level (Fig. S5a). A detailed list of the constructs is also provided in Table S4.

9. The authors have deleted a large portion of ZFAND3 and assumed that those domains are directly involved in the function of this Zinc Finger protein. However, we cannot be sure that the phenotype observed is not simply due to the fact that those deletions dramatically impair the folding of the protein, rendering it functionally inactive.

We agree with the reviewer that we cannot formally exclude an impact of protein misfolding on the domain deletion constructs. Misfolding may indeed occur in the double deletion mutant, where we also found that translocation to the nucleus was lost. However in the case of single domain deletions, the protein correctly translocates to the nucleus, suggesting that the protein is properly folded to be transported. Functional studies of other members of the ZFAND protein family (with A20 and AN1 zinc finger domains) employing similar deletions have shown that proteins with deletion of one of the two domains retain certain functions. E.g. deletion of the A20 domain in ZFAND5 still enabled binding to the proteasome, whereas deletion of the AN1 domain retained binding to polyubiquitin (Hishiya, Iemura et al. 2006). Similarly deletion of AN1 in LmSAP retained its ability to confer increased salt stress resistance (Ben Saad, Safi et al. 2019). Taken together these data support the notion that with single domain deletions, the loss of function phenotype is

likely due to the deleted domain rather than to misfolding. Nevertheless since we cannot formally rule out the hypothesis of misfolding, we have softened the conclusion in the Results and discuss this aspect in the Discussion (pages 7, 9-10)

In order to further address this point, we decided to introduce missense mutations in the Zn complexing amino acids of both domains by converting two of the Cysteine residues to Alanine in each domain (constructs M1, M2, M1-M2, see Fig. S5a) similar to what was reported for ZFAND5 (Lee, Takayama et al. 2018). Thereby we hoped to minimize the effect of the mutations on overall protein folding, while still affecting the respective Zinc finger domains. The equivalent mutation in the A20 domain of ZFAND5 (M1) abolished its ubiquitin-binding activity (Hishiya, Iemura et al. 2006, Lee, Takayama et al. 2018), whereas the missense mutations in the AN1 domain (M2) resulted in a loss of its ability to stimulate the proteasome (Lee et al., 2018). In our hands ZFAND3 with either mutated domain (M1 or M2) or both (M1-M2) did not diminish its ability to promote invasion (Fig. S5b-f), suggesting that these specific residues are not relevant for the observed ZFAND3 activity. This is in contrast to our observations with the domain deletions. All three new mutant proteins retained nuclear localization (Fig. S5g). Taken together, our data point to a requirement of the ZF domains, although we cannot conclude on the specific residues essential to confer invasion-related ZFAND3 activity. This would require a detailed mutation mapping along the protein which is beyond the scope of the present work. These data have been included and are now discussed in the result and discussion (pages 7, 9-10).

It should be noted that the concern about protein misfolding applies to any introduced mutation (even point mutations), and e.g. also to the mutated NLS with several amino acid changes. Additionally these changes might affect NLS-independent functions. In order to strengthen the notion that the observed loss of function of the NLS-mutant protein is indeed due to the changed location of the ZFAND3 protein, we successfully restored its nuclear localisation by adding a c-Myc NLS sequence (PAAKRVKLDG) to the C-terminus of the ZFAND3-mutNLS protein. This rescued the protein functionality with regard to invasion (Fig. 5a-g), clearly demonstrating the requirement of nuclear localization for ZFAND3 activity in invasion (text page 6).

10. The authors have hypothesized that ZFAND3 acts as a transcription factor within a complex (Figure 7). However, this finding is based on in silico data that were not all verified experimentally. To ensure the robustness of the data, it would be necessary to perform additional verification. The promoter sequence of COL6A2, NRCAM and FN1 harbor more than just the ZBED binding site. To ensure that the function of ZFAND3 as a transcription factor really depends on ZBED sequences, these should be mutated and no luciferase signal should be observed. Overexpression of ZFAND3 and simultaneous knockout of ZBED (1-4) should not increase the luciferase signal if the hypothesis is true. ChipSeq could be performed to ensure that ZFAND3 is indeed found at the promoter of the above-mentioned genes. The authors identified/suggested several binding partners of ZFAND3. These results must be verified by co-immunoprecipitation.

We thank the reviewer for these suggestions, which we tried to address as good as possible. We agree that the in silico predicted ZBED sequences are not the only possible interaction sites for transcriptional regulators. ZBED sites themselves are characterized by a high GC content and in close proximity numerous consensus sites for zinc finger containing DNA-binding sites sharing this feature were detected in the discussed promoter regions. We have reformulated this now in the text and in the figure to avoid the focus on ZBED (Fig. 7a-c). Following the Reviewer's suggestion, we have mutated the available GC rich regions on the promoter of the target genes (*COL6A2*, *NRCAM*, *FN1*) and found a tendency to a reduced signal in the luciferase assays (Fig. 7d-f). Since the signal did not return to control level, this is in line with the hypothesis that additional binding sites and/or additional proteins may participate in ZFAND3 activity. Since we

have introduced only 2 point mutations per binding site and not fully ablated the binding sites, residual ZFAND3 binding cannot be ruled out. These data are now included in Fig. 7d-f and text pages 8-9.

To further address the role of ZBED, we aimed to knock down ZBED4 using siRNAs as suggested by the Reviewer. Unfortunately, we were unable to achieve a relevant KD of ZBED4 using 3 different siRNA sequences both at mRNA and protein level (please see Western blots below of HEK293T cells). Control siRNAs against Cdc42 (lanes on the right) were used as a positive control for the KD experiment, while GAPDH and laminB1 were used as housekeeping proteins. We therefore were unable to conclude on this experiment.

To test for binding of ZFAND3 to the promoter region of the target genes, we performed ChIP-qPCR on NI cells overexpressing ZFAND3-FLAG and on FLAG control cells. This allowed to confirm direct binding of ZFAND3 to the promoter regions of the target genes (Fig. 7 g-i and text pages 8-9).

At the protein level, it should be noted that our BioID experiment was validated by Co-IP based on mass spectrometry analysis. Co-IP by mass spec is independent of antibody affinity and availability and is thus considered to be at least equally powerful as conventional Co-IP by Western blot (Aebersold, Burlingame et al. 2013). The validated data are presented in Fig. 6l. As requested by the Reviewer, we have further attempted to perform Co-IP by Western blot for PUF60 (*PUF60*), Pontin (*RUVBL1*) and Treacle (*TCOF1*). This allowed to validate the interaction between ZFAND3 and PUF60, which is shown on Fig. S6h. For the other two proteins, the Co-IP was not feasible for technical reasons, i.e. for Treacle the protein ran at the same size as the IgG control, while the Pontin antibody resulted in unspecific binding (multiple bands) on WB. We could not perform Co-IP on endogenous proteins, since no IP grade antibodies for ZFAND3 or ZBED4 were available. These data are included in the text page 8. To remain on the safe side, we have left out these proteins from the conclusion and only incorporated PUF60 as validated interaction partner in the schematic of protein interaction (Fig. 7j).

Reviewer #3 (Remarks to the Author):

In this report, Schuster et al. identify ZFAND3 as a candidate driver of tumor invasion in glioblastoma (GBM) using an RNA interference screen, and then perform in-depth analysis using

complementary *in vitro*, *ex vivo*, and *in vivo* approaches to validate the invasive capacity of this novel protein in patient-derived glioblastoma stem cell-like models. By manipulating ZFAND3's zinc-finger domains and nuclear localization sequence and using reporter assays, the authors go further to show that this protein is active in the nucleus and that it regulates the expression of three invasion-related genes – COL6, FN1, and NRCAM.

The drivers of tumor invasion are still poorly understood in glioblastoma, and harnessing new targets for anti-invasive therapy is an important goal in the field. The study addresses this important question in an elegant way by using an unbiased screen in a relevant model for tumor invasion to identify a target, which is then thoroughly validated by several complementary approaches in patient-derived cells.

The text is overall well written and representative of the literature in the field (with some suggestions on recent invasion literature provided below) and the experiments and conclusions are appropriately interpreted.

There are several important strengths to this study. (1) The separation into invasion-competent and invasion-defective groups provides a sound model to screen for invasion drivers. (2) The selection criterion of the 17 invasion-essential candidates is rigorous, using four different computational methods. (3) The complementary *in vivo*, *ex vivo*, and *in vivo* tools used recapitulate important aspects of GBM tumor cell invasion biology, and at least the *in vitro* analysis also accounts for aspects of tumor cell proliferation. (4) Figure 3 provides strong evidence that knockdown of ZFAND3 in highly invasive (HI) GBM cells impairs tumor migration *in vitro*, *ex vivo*, and *in vivo*. As a complement, Figure 4 also shows that overexpression of ZFAND3 in non-invasive (NI) GBM cells confers invasive potential. Importantly, the author shows that there is no effect on cell proliferation (*in vitro*). Furthermore, Figure 5 provides intriguing insight into the need for both zinc finger domains and the nuclear localization signal for pro-invasive ZFAND3 activity. A possible caveat in the results is the fact that single deletion of the zinc finger domains did not abolish nuclear translocation, yet it showed an anti-invasive phenotype, but the authors appropriately discussed the possibility for conformational change causing loss of DNA and/or protein binding as a possibility for the observed downstream effect on transcriptional activity and invasion.

My main concern is in regards to the relevance of the immunofluorescence ZFAND3 analysis in primary human samples presented in Figures 2 and S3 and its relationship to the overall message of the paper. Additional more minor concerns are outlined as well. I believe these concerns can be addressed without major revisions to this manuscript. My specific comments are provided below.

Major Concern:

1. Figure 2f-h and Supplementary Figure 3c. The immunofluorescence analysis of ZFAND3 in patient sample tissue is confusing and it does not provide very strong support for the importance of this molecule as a nuclear (transcriptional) regulator of tumor invasion. First, only rare tumor cells appear to stain positive for ZFAND3 in 2f and S3c (most P53+ tumor cells are ZFAND3-) and there are arguably more ZFAND3+P53- than ZFAND3+P53+ cells in the classifier). Second, the staining pattern of ZFAND3+ cells appears largely cytoplasmic rather than nuclear, in contrast to what is seen by later immunocytochemistry staining in NI GBM cell lines. This raises several questions that need to be reconciled in the paper.

1) What is the relevance of ZFAND3 expression in human samples if less than 1% of tumor cells show convincing protein expression? Perhaps analysis of ZFAND3 staining in a larger number of

GBM samples would be more informative, such as within a tissue microarray that includes both the core and the infiltrative tumor components. 2) ZFAND3 immunostaining appears predominantly cytoplasmic – how do the authors reconcile that with ZFAND3 activity? Perhaps analysis of nuclear vs. cytoplasmic expression of ZFAND3 in GBM patient tissue samples, in the central vs. intermediate vs. periphery zones, could be a better predictor of activity than the current analysis of ZFAND3 overall intensity in Figure 2h? 3) Are ZFAND3+P53- cells considered non-tumoral? Is ZFAND3 expressed in a subset of normal cells in both human and xenograft models (see also minor comment 4)?

We thank the Reviewer for the overall positive evaluation of our manuscript and for the important suggestions with regard to ZFAND3 expression in clinical samples, which prompted us to revisit the data on human GBM.

@ 1-2) We would like to highlight that all IHC data on core and infiltrative compartments is based on 21 human GBMs for which representative pictures are shown in Fig. 2. When we initially did these experiments we did not pay special attention to nuclear versus cytoplasmic staining and analysed the cells only based on ZFAND3 positivity using a very stringent classifier. The software-based classifier has now been adapted in order to distinguish between subcellular ZFAND3 localization in P53+/ZFAND3+ cells. The initial classifier defined a cell to express ZFAND3 if 50% of the total nuclear DAPI area was covered by ZFAND3. This cut-off has been adjusted, and a P53+ cell is now defined as having nuclear ZFAND3 expression if 10% of the nuclear P53 area is covered by ZFAND3. Cytoplasmic ZFAND3 expression is defined as ZFAND3 localization within a 4 µm perimeter from the outer edge of the P53+ nucleus while simultaneously not overlapping with P53 staining. Images showing specific nuclear ZFAND3 localization in P53+ tumor cells have been added to Figure 2f and the image of the classifier has been replaced with a more representative image (new Fig. S3a-b).

Based on these parameters, the fraction of P53+ tumor cells with ZFAND3 expression ranges from approximately 6-12%, and fractions have been calculated for both P53+/nuclear ZFAND3+ and P53+/Cytoplasmic ZFAND3+ as requested by the Reviewer. As shown earlier for the total number of ZFAND3+ cells, there was no difference in the fraction of positive cells between different tumor compartments, neither for cytoplasmic nor for nuclear staining (new Fig. S3c-d). In addition, the ZFAND3 staining intensities have been measured specifically for nuclear and cytoplasmic ZFAND3 staining. We found that both the ratio of nuclear/cytoplasmic ZFAND3 staining and the nuclear ZFAND3 staining intensity increased in the peripheral tumor cells compared to central cells, showing that the relative fraction of tumor cells with nuclear ZFAND3 expression is higher in the tumor periphery, and that nuclear ZFAND3 is expressed to a higher extent in these cells (new Fig. 2g-i and text page 5).

@ 3) This is a critical question. Although our GBM cohort represents patients with *TP53* mutations, there is heterogeneity in the tumor and not all tumor cells do express P53. We choose this marker in order to undoubtedly identify ZFAND3 staining in tumor cells, however, a portion of ZFAND3+/P53- cells likely also represents tumor cells, meaning that the fraction of ZFAND3+ tumor cells may well be underestimated. To elucidate whether ZFAND3 is expressed in non-tumor cells, we performed double-immunofluorescent stainings for Iba1/ZFAND3 (microglia/macrophages) and NeuN/ZFAND3 (neurons) in the patient samples and our xenografts (see also point 4 below). A small fraction of Iba1+ cells showed co-expression of ZFAND3, indicating that microglia/macrophages can express ZFAND3, as illustrated in Figure S3e. Neurons did not show ZFAND3 expression (Fig. S3e). These findings are now also discussed in the results (page 5). Similar data have also been obtained in xenografts (see point 4 below).

Minor Concerns:

1. Introduction. Line 71: consider including the recent CRISPR-Cas9 genetic screen study that also aims to discover new drivers of glioblastoma invasion (PMID: 31570734); and Line 68: consider mentioning recent studies exposing other transcriptional and cytoskeletal regulators (and potential therapeutic targets) of tumor migration/invasion in glioblastoma (PMID: 28122245, 30275445, 31235578, others).

We thank the Reviewer for highlighting these studies and have included them in the introduction.

2. Figure 1a: The endpoints for NI, LI, and HI seem to be quite different. The endpoint for NI in this figure is 5 weeks, while in later experiments (Figure 4k-l for example), the xenografts generated from NI cells had an endpoint of 8 weeks. The much shorter endpoint for NI vs. HI may be skewing the observation that NI tumors are non-infiltrative; a later endpoint for NI should be considered.

We are grateful to the Reviewer for pointing this out and apologize for the inadvertency. The endpoint corresponds to the tumor development time in mice, meaning the time of sacrifice due to disease and neurological symptoms. For NI cells this is indeed about 5 weeks as stated in Fig. 1, while the analysis in Fig. 4 was done at 4 weeks, which was wrongly stated in the legend. We have corrected the legend of Fig. 4.

3. Supplementary Figure 2b-d. The screen appears to only show two candidates differentially enriched in HI cells: CSF, regarded as positive control, and ZFAND3. While this does not downplay the findings in this report, some mention in the results and / or discussion in regards to this is warranted.

The expression analysis was one of the selection criteria we applied to reduce the number of candidates and thus helped to identify ZFAND3. We have clarified this in the results section (page 4). However, it does not allow to conclude much on the other candidate genes.

4. Figure 2d. In the NI and LI xenografts, some of the ZFAND3+ cells appear to show neuronal and/or microglial morphology while in the HI tissues, they appear more tumoral. It could be helpful to perform immunofluorescence co-labeling with a tumor-specific marker (human Vimentin, human nuclear antigen) and neuronal or microglial markers to define better the tumor-specific contribution of ZFAND3+ staining in these models.

Following the Reviewer's suggestion, and in addition to the double-immunofluorescence in patient samples discussed above, we have now also performed IF co-labeling for Iba1 and NeuN in the HI xenografts. In line with the patient samples, we do see some microglia/macrophages which display ZFAND3 positivity, but no double positive NeuN/ZFAND3 cells. These data have been included as Fig. S3f. It is also noteworthy that ZFAND3 staining appears to be minimal in the neuropil and is clearly enriched in the tumor area (section with NeuN staining at tumor edge).

4. Figure 4k-l. This is a very important finding in the paper and it would be helpful to visualize how the cells were scored by providing an actual annotated image of cells counted at the infiltrative margin / intermediate zone.

To clarify this point, we have now added pictures of a representative whole brain section both for Fig. 3k (HI xenografts with ZFAND3 KD) and Fig. 4k (NI xenografts with ZFAND3 overexpression) (Fig. S4j and S4k). For HI xenografts, invading cells were only counted in the contralateral hemisphere, while for NI xenografts, which grow as a circumscribed tumor, cells at the tumor border were counted. This is now better defined in the M&M section (page 13) and an annotated image of counted cells has been added (Fig. S4i).

5. Figures 3+4. The authors make a point to show that ZFAND3 does not affect cell proliferation *in vitro*, but there are no experiments assessing the effect of ZFAND3 knockdown on cell proliferation *in vivo*, which could be easily accomplished by immunostaining for an accepted cell proliferation marker in the xenograft histological sections.

We agree with the Reviewer that this would be a relevant information to add. Based on the overall tumor growth (as illustrated in Fig. S4j-k) and the Ki67 staining obtained (see below), available evidence suggests that there is no effect on proliferation *in vivo*. We have tried hard to do Ki67 stainings on our HI xenografts with ZFAND3 KD (using different antibodies), but unfortunately we were unable to obtain satisfying results. This is likely due to the fact that from these experiments we only had frozen cryostat sections available (rather than paraffin blocks) and because of the long storage period (>2 years), we must assume that tissue integrity was lost. An example is shown below, where it can be appreciated that Ki67 positivity is often filamentous (arrows) and not overlapping with DAPI. We felt that the poor quality of the staining did not allow to quantify these data and therefore did not include them in the manuscript.

6. Sample number (n) is missing from several figure legends and should be added for completion. For example, Figure 6a-b legend does not indicate how many samples were analyzed for the differential expression RNA-seq analysis, although in the methods section it was indicated that samples were sequenced in triplicates.

Where missing, we have added sample numbers in the legends of the figures. As indicated in the methods section, the RNA-Seq data was based on 3 biological replicates which is now also specified in the legend of Fig. 6.

7. Figure S3. The authors may want to also check ZFAND3 expression in the IVY GAP database, which is annotated by core/infiltrative anatomical regions.

We thank the Reviewer for this interesting suggestion. We looked up the expression of ZFAND3 in the IVY Gap database and found that ZFAND3 was heterogeneously detected in all regions. We did not detect a correlation with certain tumor regions (e.g. leading or infiltrating edge). This is however not very surprising, since the edge regions contain a large amount of non-tumor cells which dilute the signal from the remaining tumor cells. Therefore such a bulk expression analysis as presented in the IVY GAP database does not easily allow to conclude on the level of expression per region nor on the expression within specific cell types.

8. Line 107: Essential misspelled on line 107.

9. Line 171: Should be CDH2 instead of CDH1 (CDH1 is E-cadherin).

We thank the Reviewer for spotting the typos, they have been corrected.

References

Aebersold, R., A. L. Burlingame and R. A. Bradshaw (2013). "Western blots versus selected reaction monitoring assays: time to turn the tables?" Mol Cell Proteomics **12**(9): 2381-2382.

Ben Saad, R., H. Safi, A. Ben Hsouna, F. Brini and W. Ben Romdhane (2019). "Functional domain analysis of LmSAP protein reveals the crucial role of the zinc-finger A20 domain in abiotic stress tolerance." Protoplasma **256**(5): 1333-1344.

Dirkse, A., A. Golebiewska, T. Buder, P. V. Nazarov, A. Muller, S. Poovathingal, N. H. C. Brons, S. Leite, N. Sauvageot, D. Sarkisjan, M. Seyfrid, S. Fritah, D. Stieber, A. Michelucci, F. Hertel, C. Herold-Mende, F. Azuaje, A. Skupin, R. Bjerkvig, A. Deutsch, A. Voss-Bohme and S. P. Niclou (2019). "Stem cell-associated heterogeneity in Glioblastoma results from intrinsic tumor plasticity shaped by the microenvironment." Nat Commun **10**(1): 1787.

Hishiya, A., S. Iemura, T. Natsume, S. Takayama, K. Ikeda and K. Watanabe (2006). "A novel ubiquitin-binding protein ZNF216 functioning in muscle atrophy." EMBO J **25**(3): 554-564.

Lee, D., S. Takayama and A. L. Goldberg (2018). "ZFAND5/ZNF216 is an activator of the 26S proteasome that stimulates overall protein degradation." Proc Natl Acad Sci U S A **115**(41): E9550-E9559.

Meiser, J., A. Schuster, M. Pietzke, J. Vande Voorde, D. Athineos, K. Oizel, G. Burgos-Barragan, N. Wit, S. Dhayade, J. P. Morton, E. Dornier, D. Sumpton, G. M. Mackay, K. Blyth, K. J. Patel, S. P. Niclou and A. Vazquez (2018). "Increased formate overflow is a hallmark of oxidative cancer." Nat Commun **9**(1): 1368.

Reviewers' Comments:

Reviewer #2:

Remarks to the Author:

Schuster and colleagues have addressed the concerns I raised in the first review and thereby improved their manuscript. Their answers to my comments were satisfying for the most. I do not have additional comments to their manuscript.

Reviewer #3:

Remarks to the Author:

The authors have addressed all of my concerns and their manuscript is now deemed acceptable for publication.

Thank you - Nadejda Tsankova (reviewer 3)